# HIGHER-ORDER FOURIER NEURAL OPERATOR: EXPLICIT MODE MIXER FOR NONLINEAR PDES

## ABSTRACT

Neural operators provide resolution-equivariant deep learning models for learning mappings between function spaces. Among them, the Fourier Neural Operator (FNO) is particularly effective: its spectral convolution combines a low-dimensional Fourier representation with strong empirical performance, enabling generalization across resolutions. While this design aligns with settings where the Fourier basis diagonalizes the underlying operator, such as linear, constant-coefficient PDEs on periodic domains, in which Fourier modes evolve independently, nonlinear PDEs exhibit structured interactions between modes governed by polynomial nonlinearities. To capture this inductive bias, we introduce the **Higher-Order Spectral Convolution**, a spectral mixer that extends FNO from diagonal modulation to explicit $n$-linear mode mixing aligned with nonlinear PDE dynamics. Across benchmarks, including Burgers and Navier-Stokes equations, our method consistently improves accuracy in nonlinear regimes, achieving lower error while retaining the efficiency of FFT-based architectures.

## 1 INTRODUCTION

Partial differential equations (PDEs) serve as the fundamental tools for expressing the evolution of physical and engineering processes in space and time. Accurate modeling of PDE-governed systems is fundamental to understanding phenomena such as fluid dynamics (Burgers equation, Navier-Stokes equation), transport phenomena (diffusion-reaction equation) and large-scale atmospheric modeling (Shallow Water equation) (Staniforth, 2022).

For most of these equations, closed-form solutions are not available, making numerical approximation necessary. Over the past century, traditional numerical methods such as the finite difference method (FDM) (LeVeque, 2007), the finite element method (FEM) (Johnson, 1994) and the finite volume method (FVM) (LeVeque, 2002) have achieved both accuracy and interpretability, owing to their foundation in fundamental physical principles. Despite their strengths, these methods face two main limitations: high computational cost from fine spatiotemporal discretization, and reliance on full knowledge of the governing PDEs.

Therefore, in recent years, motivated by the remarkable achievements of deep learning for modeling complex functions, numerous data-driven PDE solvers have been introduced to overcome the limitations of traditional numerical methods. Among these approaches, the framework of operator learning (Kovachki et al., 2023; Berner et al., 2025) stands out as the most physically grounded. Neural operators, in particular, aim to approximate the underlying solution operator that maps input functions, such as coefficients, forcing terms, or initial conditions, to output solutions, thereby providing an approximate resolution-equivariant and efficient alternative to classical discretization-based schemes.

Among them, the Fourier Neural Operator (FNO) (Li et al., 2020), inspired by spectral methods that provide the highest spatial accuracy and exponential convergence on regular grids, stands out for modeling dynamical systems on equally spaced meshes and for its ability to transfer across resolutions without retraining, a consequence of its explicit representation in the Fourier basis, which remains consistent under mesh refinement. For complex geometries, several variants of FNO have been introduced by changing the spectral basis, for instance, the Spherical Fourier Neural Operator (SFNO) (Bonev et al., 2023) on the sphere, and NORM (Chen et al., 2023) on general Riemannian manifolds. Furthermore, extensions to irregular meshes have been proposed by mapping them onto

regular grids, either via a general learnable map (GNO (Li et al., 2023b)), a learnable diffeomorphism (GEO-FNO (Li et al., 2023a)), or an optimal transport map (OTNO (Li et al., 2025)).

In this work we will refer to this class of models as *spectral neural operators* (SNOs) due to their explicit modeling of the spectrum of modes of the input function, with a classic or generalized Fourier transform, with and without encoders and decoders. SNOs are typically composed of linear layers and nonlinear activation functions. The linear components are usually global convolutions over a truncated set of modes, and they evolve Fourier modes independently, without mixing. To augment the approximation power of the SNO layers we propose an $n$-order spectral convolution that implements a $n$-linear global mixing of Fourier coefficients while retaining the computational efficiency of a Fourier truncation.

The spectral convolution of a SNO closely mimics the action of the Green function, a kernel whose convolution yields the solution of linear PDEs with constant coefficients on periodic domains (Stakgold & Holst, 2011). For nonlinear PDEs, variable coefficients or non-periodic geometries, the Green function no longer provides a useful representation, yet the composition of linear spectral convolutions with nonlinear activations endows SNOs with universal approximation capabilities (Kovachki et al., 2021).

Much like SNOs, 2-layer MLPs also enjoy universal approximation properties (Cybenko, 1989; Chen & Chen, 1996). However, modern deep learning has highlighted the advantages of richer nonlinear layers, most notably the attention mechanism (Bahdanau, 2014; Vaswani et al., 2017). Transformer models have rapidly become the dominant architecture across various application domains, spanning language, vision (Dosovitskiy et al., 2020), chemistry (Jumper et al., 2021), and more recently physical modeling (Alkin et al., 2024; Colagrande et al., 2025). A key factor behind their success is the ability of classical attention to capture pairwise interactions in physical space. This mechanism has recently been generalized to model interactions among an arbitrary number $n$ of entities, giving rise to higher-order attention (Clift et al., 2019).

Despite their $O(\texttt{seq\_len}^n)$ complexity in the sequence length, these higher-order variants show better scaling laws (Roy et al., 2025) and exponentially improved depth efficiency on dedicated tasks (Sanford et al., 2023). Following this line of work, we introduce a new framework that realizes $n$-order interactions between coefficients directly in the Fourier domain, providing the spectral analogue of higher-order attention, which operates in the Dirac domain. Crucially, our method avoids the $O(\texttt{seq\_len}^n)$ blow-up of higher-order attention and matches FFT-based SNOs with a complexity of $O(\texttt{seq\_len} \log(\texttt{seq\_len}))$ per layer.

More similar to our work are the triangular attention mechanism of the edge transformer (Bergen et al., 2021) and the triangle attention of AlphaFold2 (Jumper et al., 2021). In both cases, the triangle refers to three-way interactions in the spatial domain: given a triplet of nodes, triangular attention models the dependencies along the edges of the corresponding triangle, enabling richer geometric reasoning. In contrast, our triadic (order $n = 2$) spectral convolution realizes the analogue of this mechanism in the Fourier domain: the triangle here corresponds to a triplet of frequency modes whose wavevectors satisfy a closure relation (e.g. $k_1 + k_2 = k_3$), capturing the nonlinear triadic interactions that govern energy transfer in PDE dynamics. For $n > 2$, our method can be viewed as the Fourier analogue of a $n$-symplicial extension of the aforementioned attention mechanisms.

On the neural operator side, the Dynamic Schwartz–Fourier Neural Operator (DSFNO) Gao et al. (2025), has recently been introduced to address the limitations of the static convolution kernel used in FNO. DSFNO employs spectral convolutions whose kernels are dynamically generated, via a hyper-network, from truncated activations, leading to improved performance. However, the resulting kernels remain largely unstructured. In this work, we extend this line of research by explicitly structuring the kernel to match the interaction patterns dictated by the solution operators of polynomial nonlinear PDEs.

Our contributions are the following:

1. **Higher-Order Fourier Neural Operators.** We design the first spectral neural operators modeling the exact mode interaction of non-linear PDEs.

2. **Interaction on different geometries.** We showcase the effect of modeling order 2 interactions on spherical data by applying our method to Spherical Harmonic convolutions.

3. **Experiments and ablation studies.** Through extensive experiments, we show the advantages of the proposed design in non-linear settings.

## 2 SETTING AND NOTATION

We consider a time-dependent PDE defined on a spatial domain $\Omega \subset \mathbb{R}^d$, with boundary $\partial\Omega$, $d$ the number of spatial dimensions, and temporal domain $[0, T]$. A solution $u(x, t)$ of this PDE satisfies the general system described in Eq. 1, where $F$ is a function of the solution $u$ and of its spatial derivatives $\frac{\partial^i u}{\partial x_i}$, $\nu$ represents a set of PDE coefficients, $\mathcal{B}$ encodes the boundary conditions, and $u^0$ denotes the initial condition sampled from a probability distribution on $L_2(\Omega, \mathbb{R})$, i.e. $u^0 \sim p^0(\cdot)$.

$$
\begin{aligned}
\frac{\partial u}{\partial t} &= F\left(\nu, t, x, u, \frac{\partial u}{\partial x}, \frac{\partial^2 u}{\partial x_2}, \dots\right), &&\forall x \in \Omega, \ \forall t \in (0, T], \\
\mathcal{B}(u)(t, x) &= 0, &&\forall x \in \partial\Omega, \ \forall t \in (0, T], \\
u(0, x) &= u^0(x), &&\forall x \in \Omega.
\end{aligned}
\tag{1}
$$

The operator learning task we consider consists in predicting the solution operator $\mathcal{G}$, defined in Eq. 2, that propagates the physical state one time step forward:

$$
\begin{aligned}
\mathcal{G} : L^2(\Omega, \mathbb{R}) &\to L^2(\Omega, \mathbb{R}) \\
u(\cdot, t) &\mapsto u(\cdot, t+1)
\end{aligned}
\tag{2}
$$

**Polynomial nonlinearities in PDEs.** We can write Eq. 1 as follows in Eq. 3 by aggregating its terms based on the degree of nonlinearity:

$$
\frac{\partial u}{\partial t} = \sum_{n \in \mathbb{N}} \mathsf{P}_{\mathcal{I}, n}(u(x, t))
\tag{3}
$$

where $\mathsf{P}_{\mathcal{I}, n}(u(x, t))$ contains the $n$-linear components of the PDE and it is a homogeneous polynomial in the partial derivatives $\frac{\partial^i u}{\partial x_i}(x, t)$ and $\mathcal{I}$ is the set of multi-indices $\alpha = (\alpha_1, \dots, \alpha_n)$ of the partial derivatives in each monomial $\prod_{i=1}^n \frac{\partial^{\alpha_i} u}{\partial u_{\alpha_i}}$. We refer to the maximal value of $n$ as the *degree of nonlinearity* of the PDE. We now focus on the $n$-linear part of the equation

$$
\mathsf{P}_{\mathcal{I}, n} = \sum_{\alpha \in \mathcal{I}} c_\alpha \prod_{i=1}^n \frac{\partial^{\alpha_i} u}{\partial x_{\alpha_i}}
\tag{4}
$$

For our analysis we consider functions defined on the torus, i.e. $\Omega = \mathbb{T}^d$, and we restrict to scalar functions, i.e. $u : \mathbb{T}^d \to \mathbb{R}$. In the periodic setting it is convenient to expand $u$ in Fourier basis as in Eq. 5.

$$
u(x, t) = \sum_{k \in \mathbb{Z}^d} \widehat{u}(k, t) e^{ik \cdot x}, \quad \widehat{u}(k, t) \in \mathbb{C}.
\tag{5}
$$

Therefore we consider the Fourier transform of the $n$-linear part of the PDE in Eq. 4:

$$
\mathsf{P}_{\mathcal{I}, n}(\widehat{u})(k, t) = \sum_{k_1 + \dots + k_n = k} C \, \widehat{u}(k_1, t) \widehat{u}(k_2, t) \cdots \widehat{u}(k_n, t)
\tag{6}
$$

Where $C = C(k, \alpha)$ is a constant dependent on the multi-index $\alpha$ and the index $k$.

The summation term of Eq. 6 corresponds to the $n$-linear convolution of Fourier modes. It captures how input frequencies combine under the nonlinearity, and it is precisely this mixing that our higher-order spectral convolution is designed to model, with $C(k, \alpha)$ providing the learnable kernel.

Quadratic interactions ($n = 2$) appear in Burgers, in the Navier-Stokes equations and in the rotated, hyperviscous, forced Shallow Water Equations on the sphere while cubic non-linearities ($n = 3$) appear in the Diffusion-reaction equation.

We refer to the appendix B for a more detailed discussion and present in section 3 the explicit construction in the case of Navier-Stokes equations.

## 3 A CONCRETE EXAMPLE: NAVIER-STOKES EQUATIONS

We present here, as example, the non-linear interactions on the incompressible Navier-Stokes equation that is usually written as follows in Equation 7 in the velocity form.

$$\partial_t w(x,t) + u(x,t) \cdot \nabla w(x,t) = \nu \Delta w(x,t) + f(x) \qquad x \in (0,1)^2,\ t \in (0,T] \qquad (7)$$

$$\nabla \cdot u(x,t) = 0 \qquad x \in (0,1)^2,\ t \in [0,T] \qquad (8)$$

$$w(x,0) = w_0(x) \qquad x \in (0,1)^2 \qquad (9)$$

The task typically requires to predict the evolution of the vorticity $w$ (Li et al., 2020) (Serrano et al., 2024) so we express the PDE in the vorticity form as follows in Equation 10:

$$\partial_t(w) = \nu \Delta w(x,t) - (\nabla^\perp \Delta^{-1} w) \cdot \nabla w(x,t) + f(x) \qquad x \in (0,1)^2,\ t \in (0,T] \qquad (10)$$

$$\nabla \cdot \nabla^\top \Delta^{-1} w = 0 \qquad x \in (0,1)^2,\ t \in [0,T] \qquad (11)$$

$$w(x,0) = w_0(x) \qquad x \in (0,1)^2 \qquad (12)$$

To observe the interaction of the Fourier modes of the vorticity we take the Fourier transform, for $k \in \mathbb{Z}^2,\ t \in (0,T]$:

$$\partial_t(\widehat{w})(k,t) = -\nu(2\pi)^2 |k|^2 \widehat{w}(k,t) - \sum_{p+q=k} \frac{(p+q) \cdot p^\perp}{|p|^2} \widehat{w}(p,t)\widehat{w}(q,t) + \widehat{f}(k,t). \qquad (13)$$

In Fourier space, the nonlinear advection term in the Navier-Stokes equations becomes a convolution integral, and a triad interaction term in the turbulence kinetic energy equation. Despite being conservative, and therefore contributing only to energy exchange between Fourier modes, this term is at the heart of many of the interesting questions in the literature.

As highlighted in (Cheung & Zaki, 2014), the primary difficulty in working with the spectral Navier–Stokes equations described in Eq.13, is to appropriately account for all nonlinear interactions. An analytical treatment requires some means of tracking energy transfer from two arbitrary modes $p$ and $q$ into a third mode $k$. Therefore, it motivates the use of architectures that go beyond diagonal modulation of Fourier coefficients by explicitly parameterizing higher-order interactions in the spectral domain. Note that quadratic nonlinearity (order $n = 2$) yields triadic interactions $(p, q, k)$ with $p + q = k$; hence an order-2 corresponds to triads in spectral turbulence.

**Neural Operator.** Following the framework of (Kovachki et al., 2023), a Neural Operator $\mathcal{G}_\theta$ is implemented as a stacked structure of $L$ learnable layers $\mathcal{Q}_\ell$, inserted between point-wise neural networks denoted $\mathcal{L}$ (lifting network) and $\mathcal{P}$ (projection network) that elevate the lower-dimensional input to a higher-dimensional latent space and projects the transformed input back to a lower-dimensional output dimension, respectively.

$$\mathcal{G}_\theta = \mathcal{P} \circ \mathcal{Q}_1 \cdots \circ \mathcal{Q}_L \circ \mathcal{L} \qquad (14)$$

We denote by $v_\ell$ the hidden representation at layer $\ell$. The operator layer $\mathcal{Q}_\ell : v_\ell \mapsto v_{\ell+1}$ performs the iterative update described in Eq. 15, where $W_\ell$ is a point-wise linear map, $b_\ell$ a bias and $\mathcal{K}_\ell$ an integral operator.

$$\mathcal{Q}_\ell(v_\ell) = \sigma(W_\ell v_\ell + \mathcal{K}_\ell(v_\ell) + b_\ell) \qquad (15)$$

**Fourier Neural Operator (FNO).** FNO (Li et al., 2020) follows the framework described in section 2 and implements the integral kernels $\mathcal{K}$ as global convolution operators $\mathcal{C}$ preceded by a truncation of Fourier coefficients $\mathsf{T}_M(u)(x) = \sum_{|k| \le M} \hat{u}(k) e^{ik \cdot x}$ where $M$ is the number of retained modes. The so-called *spectral convolution* writes as follows in Eq. 16 in physical space, where $\kappa_\theta$ is a kernel parameterized by $\theta$.

$$\mathcal{C}_\theta(v)(x) = \int_\Omega \kappa_\theta(x - y) \mathsf{T}_M v(y) dy \qquad (16)$$

The Fourier Neural Operator (FNO) implements this map efficiently by parameterizing $\kappa_\theta$ in the Fourier domain, acting mode-wise, and then returning to physical space via inverse FFT:

$$\widehat{\mathcal{C}_\theta v}(k) = W_k \widehat{v}(k) \tag{17}$$

While successful on many tasks, this architecture does not explicitly capture *multi-linear frequency mixing*, since each mode is updated independently and interactions are only induced indirectly through point-wise nonlinearities between different layers.

**Higher-Order Fourier Neural Operators (HO-FNO).** We extend the kernel map to incorporate explicit $m$-linear interactions via the following Higher-Order Spectral Convolution:

$$\big(\mathcal{H}_\theta u\big)(x) = \int_\Omega k_\theta(x - y) \mathsf{T}_M\big((A_1 u)(y)(A_2 u)(y) \cdots (A_m u)(y)\big) dy \tag{18}$$

Here, each $A_i$ is a learnable linear operator acting channel-wise in physical space. In this work, we instantiate $A_i$ as per-point linear maps shared across spatial locations but not across layers. Alternative parameterizations are left to future work.

The $m$-linear point-wise products in physical space induces a structured $m$-linear global mixing among Fourier coefficients as described in Eq. 19:

$$(\widehat{\mathcal{H}_\theta v})(k) = W_k \sum_{k_1 + \ldots + k_m = k} A_1 \widehat{v}(k_1) A_2 \widehat{v}(k_2) \cdots A_m \widehat{v}(k_m) \tag{19}$$

Thus, each mode $k$ aggregates all $m$-tuples of modes with indices summing to $k$, mirroring the nonlinear interaction structure of PDEs with polynomial nonlinearities.

We emphasize that the operator is evaluated only for modes $k \leq m$, preserving the computational efficiency of FNO. However, each retained mode $k$ is updated using information from all Fourier modes, rather than being restricted to the truncated subset. We find it beneficial for training stability to normalize the multilinear terms. Table 2 reports results with and without RMS normalization, illustrating the consistent improvements obtained with this normalization scheme.

This $m$-linear convolution provides a principled, FFT-efficient mechanism for explicit mode mixing in neural operators, extending the FNO beyond purely mode-wise updates. The $m$-linear interaction can be carried out in $\mathcal{O}(N \log N)$ complexity by multiplying fields pointwise in the physical domain, transforming to Fourier space via FFT, applying spectral multipliers, and mapping back with an inverse FFT. In practice, each layer uses one FFT and one inverse FFT per channel group; $m$-linear mixing is effected via pointwise products, so the asymptotic cost remains $O(N \log N)$.

**Parameter Count** For completeness, we report the number of parameters in a single spectral layer. The standard FNO employs a complex-valued kernel with $MC^2$ parameters, where $C$ denotes the number of input and output channels, and $M$ is the total number of retained Fourier modes. Our HO-spectral convolution of order $m$ introduces $m$ additional weight matrices of size $C \times C$, shared across spatial locations, contributing $mC^2$ parameters. Thus, the total parameter count becomes

$$MC^2 + mC^2 \tag{20}$$

growing linearly with the interaction order $m$. We emphasize that $m$ is typically much smaller than the total number of retained modes $M$, making the additional parameters introduced by our higher-order blocks negligible in practice, as confirmed by the parameter counts reported in Table 2.

We summarize our proposition in Figure 1.

## 4 EXTENSION TO GENERALIZED FOURIER TRANSFORMS

The classical Fourier transform is defined for functions defined on the torus $\mathbb{T}_d$. When a function is instead defined on a manifold $\mathcal{M} \subset \mathbb{R}^D$, one can still apply the classical Fourier transform by first extending the function to the ambient euclidean space $\mathbb{R}^D$. While this procedure makes the transform computable, the resulting representation ignores the geometry of the domain $\mathcal{M}$ of the function and therefore provides a sub-optimal representation.

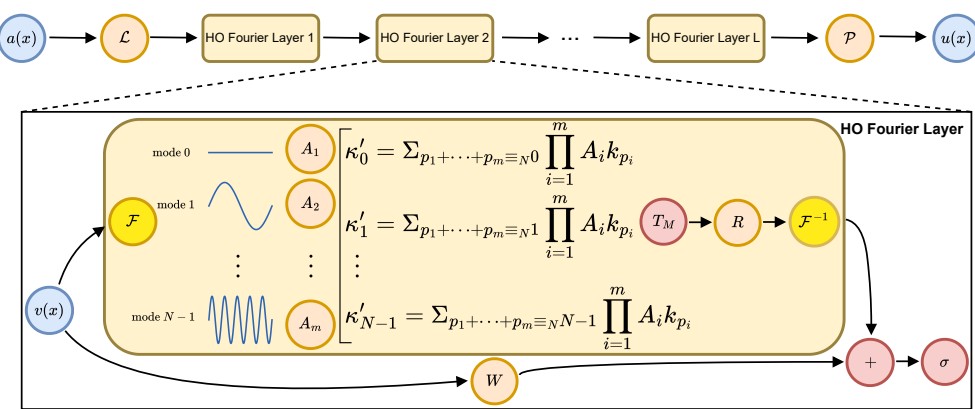

Figure 1: Overview of our proposed HO-FNO (illustration adapted from (Li et al., 2020)).
**Top: Neural operator architecture.** An input $a$ is lifted to a higher-dimensional channel space by a neural network $\mathcal{L}$. A number $L$ of HO-FNO layers are then applied to the lifted input, before it is projected back to the target dimension by a neural network $\mathcal{P}$ to obtain the output $u$.
**Bottom: High-Order Fourier layer.** An intermediary input $v$ is processed by a HO-Fourier layer. Its Fourier transform $F$ is computed, producing $N$ modes. Our method mixes these modes to obtain $N$ new pseudo-modes: $\kappa'_0, \cdots, \kappa'_{N-1}$. Here, a pseudo-mode $\kappa_i, i \in [0, N-1]$ is obtained by mixing the subset of $m$ original modes defined by $\{p_1, p_2, ...p_m \in [0, N-1], p_1+p_2+...+p_m = i \bmod N\}$. Only the $M$ lower Fourier pseudo-modes $\kappa'_1, \cdots, \kappa'_M$ are kept. HO-FNO then applies a linear transform $R$ on those $M$ lower Fourier pseudo-modes, and applies the inverse Fourier transform $F^{-1}$. Through a skipped connection, the mixed output is combined with the original input transformed by a local linear transform $W$ and a non-linear activation $\sigma$ is applied.

To overcome this limitation, the notion of a Fourier basis has been generalized to arbitrary compact Riemannian manifolds $\mathcal{M}$ through the spectral decomposition of the Laplace–Beltrami operator. Concretely, one considers the eigenvalue problem

$$-\Delta_g \phi_j = \lambda_j \phi_j \quad \text{on } \mathcal{M}, \tag{21}$$

where $\Delta_g f = \mathrm{div}_g(\nabla_g f)$ denotes the Laplacian, defined as the divergence of the Riemannian gradient. The eigenfunctions $\phi_j$ serve as generalized Fourier modes, while the corresponding eigenvalues $\lambda_j$ play the role of frequencies

For most manifolds, the eigenfunctions of the Laplace–Beltrami operator do not admit a closed-form expression and must be precomputed numerically (Chen et al., 2023). An important exception is the sphere, where the generalized Fourier modes correspond to the well-known *spherical harmonics*. This extension of the Fourier transform naturally induces a corresponding notion of convolution, defined as a linear diagonal operator in the generalized Fourier domain. In the same spirit, Higher-Order Spectral Convolutions also extend to arbitrary geometries, and the theoretical framework developed in the classical Fourier setting remains directly applicable.

We illustrate this by experimenting with the rotated, hyperviscous, forced Shallow Water Equation (SWE) on the sphere, with results reported in Table 3.

## 5 EXPERIMENTS

**Tasks.** We experiment with simulation tasks from PDEBench (Takamoto et al., 2022), namely the 1D Burger's equation with viscosity $\nu = 0.001$, and the 2D Diffusion-Reaction equation. We consider three 2D Navier-Stokes datasets with viscosity $\nu = 10^{-3}$, $\nu = 10^{-4}$ and $\nu = 10^{-5}$, provided by (Serrano et al., 2024; Li et al., 2020). In addition, we include the rotated, hyperviscous, forced Shallow Water Equation (SWE) on the sphere (McCabe et al., 2023a), made available through The Well (Ohana et al., 2024b) and we show resolution-equivariance on the Darcy Flow dataset provided in (Li et al., 2020). All datasets are used in their standard form, except for SWE, which

we subsample for shorter training (see Appendix C.4). We focus on nonlinear dynamics to better highlight the advantages of the proposed method.

**Metrics.** We evaluate models using three complementary metrics: Mean Squared Error (MSE), Normalized Mean Squared Error (NRMSE), and Rollout NRMSE. MSE captures predictive accuracy in physical space, while NRMSE rescales the error by the target norm, enabling fair comparison across datasets of different magnitudes. We also report Rollout NRMSE over full trajectories: although rollout stability is not a focus of this work, it provides useful insight into long-term performance in settings closer to real-world applications. When data are normalized for training stability, predictions are denormalized before computing the loss. We refer to Appendix D for more details on each metric.

**Baselines.** We compare our proposed HO-FNO and HO-SFNO against several representative baselines. On planar geometries, we use the original FNO (Li et al., 2020), while for data on the sphere we adopt SFNO (Bonev et al., 2023). We further include UNO (Rahman et al., 2022), a U-Net–style *neural operator* that combines encoder–decoder contractions/skip connections with Fourier-domain operator layers (as in FNO), enabling much deeper stacks at similar memory cost. Together with FNO, which tests spectral operators without multiscale contracting paths, and U-Net, which tests purely pixel-space convolution without learned spectral operators, the baselines provide a point of comparison between standard convolutional models and spectral neural operators. We also include DSFNO (Gao et al., 2025) for comparison, reporting the original results from the paper.

**Architecture.** We used models with comparable parameter counts across datasets, adjusting their size to match task difficulty while ensuring that all experiments can be trained for 100 epochs within 15 hours on a single NVIDIA A100 GPU. The resulting models contain approximately 2.3M parameters for PlanetSWE, 600K parameters for the Diffusion–Reaction equation, 80K parameters for the Burgers equation and approximately 1M parameters for Navier Stokes to ensure fair comparison with Gao et al. (2025) and Li et al. (2020).

For our baselines based on Neural Operators, we adopt linear pointwise lifting and projection networks, denoted $\mathcal{P}$ and $\mathcal{Q}$. We use 4 layers, with embedding dimension 32, except for PlanetSWE where the embedding dimension is 64. We retain 16 modes for every task except for Navier Stokes, where 22 modes in each spatial dimension is kept to ensure same size models with Gao et al. (2025) and Li et al. (2020).

In Table 1, we report our experiments on non-turbulent datasets, adding UNet and UNO baselines, for which we use the standard architecture with 4 layers and an initial embedding dimension of 12 for Burgers and 16 for Diffusion Reaction, chosen to match or exceed the parameter counts of the operator-learning models. For UNO, we retain the same number of modes as in the corresponding Neural Operator baselines for each dataset. In Table 2, we report our experiments on three Navier Stokes dataset variants with various viscosity $\nu$ and compare our HO-FNO implementation with FNO and the results presented by Gao et al. (2025).

**Hyperparameters.** The higher-order variation of the Fourier Neural Operator introduced in this work does not introduce additional hyperparameters beyond those of the standard architecture. The main hyperparameters of the models are therefore the number of layers, the latent embedding dimension per layer, and the number of retained Fourier modes in each spatial dimension (1 for Burgers and 2 for all other datasets).

All models were optimized with AdamW, for 100 epochs in Table 1 and Table 3 and for 500 epochs for Table 2 as done in Li et al. (2020) and Gao et al. (2025).

**Results.** Table 1 compares U-Net, UNO, FNO, and our proposed HO-FNO across four PDE benchmarks under three criteria: MSE , nRMSE , and rollout nRMSE . Overall, HO-FNO attains the best single-step accuracy on all datasets, consistently outperforming both FNO and UNO (the latter sometimes by modest margins, e.g., on Diffusion–Reaction).

On Burgers (1D), HO-FNO reduces MSE from $3.6 \times 10^{-6}$ to $2.4 \times 10^{-6}$ and nRMSE from $2.0 \times 10^{-3}$ to $1.6 \times 10^{-3}$, while rollout nRMSE is comparable to FNO ($8.0 \times 10^{-2}$ vs $7.5 \times 10^{-2}$).

Table 1: Test performance of different models trained on MSE. We report validation MSE , normalized RMSE ( nRMSE ), and rollout nRMSE , visualizations are provided in Appendix F. Best results per metric are in **bold**.

| Dataset | Metric | U-Net | UNO | FNO | HO-FNO (ours) |
|---|---|---|---|---|---|
| Burgers (1D) | MSE | $7.4 \times 10^{-1}$ | $3.5 \times 10^{-6}$ | $\underline{3.6 \times 10^{-6}}$ | $\mathbf{2.4 \times 10^{-6}}$ |
| | nRMSE | $3.3 \times 10^{-1}$ | $2.6 \times 10^{-3}$ | $\underline{2.0 \times 10^{-3}}$ | $\mathbf{1.6 \times 10^{-3}}$ |
| | Rollout | Diverged | $1.04$ | $\mathbf{7.5 \times 10^{-2}}$ | $\underline{8.0 \times 10^{-2}}$ |
| Diffusion-Reaction (2D) | MSE | $3.3 \times 10^{-3}$ | $\underline{8.4 \times 10^{-5}}$ | $9.2 \times 10^{-5}$ | $\mathbf{8.3 \times 10^{-5}}$ |
| | nRMSE | $2.6 \times 10^{-1}$ | $\underline{7.3 \times 10^{-2}}$ | $8.5 \times 10^{-2}$ | $\mathbf{6.7 \times 10^{-2}}$ |
| | Rollout | $\mathbf{1.01}$ | $\underline{1.59}$ | $5.28$ | $2.37$ |

Table 2: Test performance of FNO and HO-FNO variants (orders up to 3) on Navier–Stokes datasets with and without RMS Norm applied to the multilinear terms. We report the number of parameters, the validation MSE , the normalized MSE ( nRMSE ), the rollout nRMSE as well as the wall-clock time for a single-sample inference and for a training batch of size 64. An extended version with baselines taken from Gao et al. (2025) can be found in Appendix E.

| model | | FNO | HO-FNO | | | | DSFNO* |
|---|---|---|---|---|---|---|---|
| order | | 1 | 2 | | 3 | | |
| RMS Norm | | no | no | yes | no | yes | |
| N. parameters | | 1 085 729 | 1 094 177 | 1 094 177 | 1 098 401 | 1 098 401 | 1.06M |
| **NS** ($\nu = 10^{-3}$) | MSE | $3.0 \times 10^{-7}$ | $2.5 \times 10^{-7}$ | $7.8 \times 10^{-8}$ | $2.1 \times 10^{-7}$ | $\mathbf{7.6 \times 10^{-8}}$ | – |
| | nRMSE | $4.4 \times 10^{-4}$ | $4.0 \times 10^{-4}$ | $\underline{2.8 \times 10^{-4}}$ | $3.8 \times 10^{-4}$ | $\mathbf{2.7 \times 10^{-4}}$ | – |
| | Rollout | $1.2 \times 10^{-2}$ | $1.1 \times 10^{-2}$ | $\underline{1.8 \times 10^{-3}}$ | $9.7 \times 10^{-3}$ | $\mathbf{1.6 \times 10^{-3}}$ | $5.6 \times 10^{-3}$ |
| **NS** ($\nu = 10^{-4}$) | MSE | $2.6 \times 10^{-3}$ | $1.0 \times 10^{-3}$ | $\mathbf{7.9 \times 10^{-4}}$ | $9.8 \times 10^{-4}$ | $\underline{7.9 \times 10^{-4}}$ | – |
| | nRMSE | $2.9 \times 10^{-2}$ | $1.5 \times 10^{-2}$ | $\mathbf{1.3 \times 10^{-2}}$ | $1.5 \times 10^{-2}$ | $\underline{1.3 \times 10^{-2}}$ | – |
| | Rollout | $7.7 \times 10^{-2}$ | $4.8 \times 10^{-2}$ | $\mathbf{4.6 \times 10^{-2}}$ | $4.8 \times 10^{-2}$ | $\underline{4.6 \times 10^{-2}}$ | $6.0 \times 10^{-2}$ |
| **NS** ($\nu = 10^{-5}$) | MSE | $1.8 \times 10^{-2}$ | $\underline{1.7 \times 10^{-2}}$ | $\mathbf{1.7 \times 10^{-2}}$ | $1.8 \times 10^{-2}$ | $1.8 \times 10^{-2}$ | – |
| | nRMSE | $6.7 \times 10^{-2}$ | $\underline{6.5 \times 10^{-2}}$ | $\mathbf{6.5 \times 10^{-2}}$ | $6.8 \times 10^{-2}$ | $6.8 \times 10^{-2}$ | – |
| | Rollout | $1.3 \times 10^{-2}$ | $\underline{1.1 \times 10^{-2}}$ | $\mathbf{1.1 \times 10^{-2}}$ | $1.2 \times 10^{-2}$ | $1.2 \times 10^{-2}$ | – |
| Wall-clock time | inference (ms) | $1.4 \pm 0.12$ | $1.8 \pm 0.14$ | $2.2 \pm 0.22$ | $1.9 \pm 0.19$ | $2.7 \pm 0.31$ | – |
| | training (ms) | $7.91 \pm 0.34$ | $11.2 \pm 0.44$ | $16.1 \pm 0.61$ | $13.4 \pm 0.50$ | $21.9 \pm 0.70$ | – |

\* Original errors reported in Gao et al. (2025), single-step metrics and results for Navier–Stokes with $\nu = 10^{-5}$ were not provided.

On Diffusion–Reaction (2D), HO-FNO improves one-step accuracy (MSE $8.3 \times 10^{-5}$ vs $9.2 \times 10^{-5}$; nRMSE $6.7 \times 10^{-2}$ vs $8.5 \times 10^{-2}$) and substantially lowers rollout relative to FNO (2.37 vs 5.28). Notably, UNO achieves an even smaller rollout (1.59) despite weaker single-step metrics, and U-Net reports a low rollout (1.01) while being orders of magnitude worse on one-step errors, underscoring the need to interpret rollout normalization and horizon with care.

Across all Navier–Stokes settings, HO-FNO yields consistent accuracy gains over FNO without increasing parameter count, with the magnitude of improvements depending on viscosity. At $\nu = 10^{-3}$, where the dynamics are smoother, we observe the largest benefits: MSE drops from $3.0 \times 10^{-7}$ to $7.8 \times 10^{-8}$, nRMSE from $4.4 \times 10^{-4}$ to $2.8 \times 10^{-4}$, and rollout error decreases by more than an order of magnitude, from $1.2 \times 10^{-2}$ to $1.1 \times 10^{-3}$. At $\nu = 10^{-4}$, HO-FNO continues to provide substantial improvements, reducing MSE from $2.6 \times 10^{-3}$ to $7.9 \times 10^{-4}$, nRMSE from $2.9 \times 10^{-2}$ to $1.3 \times 10^{-2}$, and rollout from $7.7 \times 10^{-2}$ to $4.6 \times 10^{-2}$, outperforming DSFNO in rollout accuracy at comparable parameter count. Even in the most challenging regime $\nu = 10^{-5}$, where errors are smaller and gains are harder to achieve, HO-FNO still improves MSE (from $1.8 \times 10^{-2}$ to $1.7 \times 10^{-2}$), nRMSE (from $6.7 \times 10^{-2}$ to $6.5 \times 10^{-2}$), and rollout (from $1.3 \times 10^{-2}$ to $1.1 \times 10^{-2}$). These results indicate that explicitly modeling higher-order mode interactions consistently enhances both single-step and long-term predictions, with the largest impact in regimes where structured spectral coupling is most informative. We also observe that applying RMS Norm to the multilinear terms noticeably improves the stability of HO-FNO, leading to consistent gains in both single-step and rollout metrics, at the cost of a modest increase in runtime.

Table 3: Test performance on rotated, hyperviscous, forced Shallow Water Equation (SWE). We trained the models with MSE and report test MSE, NRMSE and Rollout NRMSE for time intervals $(0, 10)$, $(11, 25)$, $(26, 50)$ and full rollout. Best per metric in **bold**.

| Model | MSE | NRMSE | Rollout $(0:10)$ | Rollout $(11:25)$ | Rollout $(26:50)$ | Rollout |
|---|---|---|---|---|---|---|
| SFNO | 8.23 | $1.7 \times 10^{-2}$ | $9.9 \times 10^{-2}$ | $3.0 \times 10^{-1}$ | $7.2 \times 10^{-1}$ | $7.7 \times 10^{-1}$ |
| HO-SFNO (ours) | **5.56** | $\mathbf{1.3 \times 10^{-2}}$ | $\mathbf{8.0 \times 10^{-2}}$ | $\mathbf{2.6 \times 10^{-2}}$ | $\mathbf{6.2 \times 10^{-1}}$ | $\mathbf{7.0 \times 10^{-1}}$ |

The visualization of rollouts in the Appendix D (see Figures 3-9) offers a qualitative view of the stability of the simulation, and corroborates these trends. On Navier–Stokes, HO-FNO preserves coherent vortical filaments and shear layers over long horizons (e.g., see $t=10$, $t=19$), whereas UNO and U-Net seem to be unable to reconstruct an image close to the target. FNO is visually closer to HO-FNO at $\nu=10^{-4}$ and $\nu=10^{-5}$, but fails to reconstruct useful patterns in the Diffusion-Reaction equation. Overall, while UNO and U-Net can attain strong rollout performance , a visual check of their reconstructed images reveals that they are visually very far from the target, corroborating with their weaker one-step accuracy when compared to HO-FNO.

In summary, higher-order spectral mixing improves accuracy broadly and stabilizes long-horizon predictions in regimes with strong nonlinear mode coupling, with the largest relative gains on low-viscosity Navier–Stokes. UNO remains a strong baseline, particularly on Diffusion–Reaction rollout, yet HO-FNO consistently provides the best single-step accuracy and the visually strongest rollout improvements.

**Results on spherical data.** Table 3 compares the spherical baseline SFNO to our HO-SFNO on the rotated, hyperviscous, forced SWE. HO-SFNO achieves the best score on every metric: test MSE drops from $8.23$ to $\mathbf{5.56}$ and NRMSE from $1.7 \times 10^{-2}$ to $\mathbf{1.3 \times 10^{-2}}$; rollout errors are uniformly lower across horizons (e.g., early $[0, 10]$ decreases from $9.9 \times 10^{-2}$ to $\mathbf{8.0 \times 10^{-2}}$, late $[26, 50]$ from $7.2 \times 10^{-1}$ to $\mathbf{6.2 \times 10^{-1}}$), and the overall rollout NRMSE improves from $7.7 \times 10^{-1}$ to $\mathbf{7.0 \times 10^{-1}}$. These consistent gains support the inductive bias behind HO-SFNO: SWE on the sphere features quadratic wave–vortex couplings that are naturally represented in the spherical harmonic domain, and adding explicit $m$-linear spectral mixing on top of the SFNO backbone better aligns the model with these multi-mode interactions, yielding higher single-step fidelity and more stable long-horizon behavior.

**Comparison with FNO: Speed and resolution equivariance** To ensure HO-FNO maintains the resolution equivariance property of FNO, we train HO-FNO on the Darcy flow dataset at resolution $200 \times 200$ and test the obtained model at resolutions ranging from $50 \times 50$ to $400 \times 400$. The obtained results are showed Figure 2a. We also compare the speed of our proposed method to FNO and report the results for training and testing in Figure 2b, c.

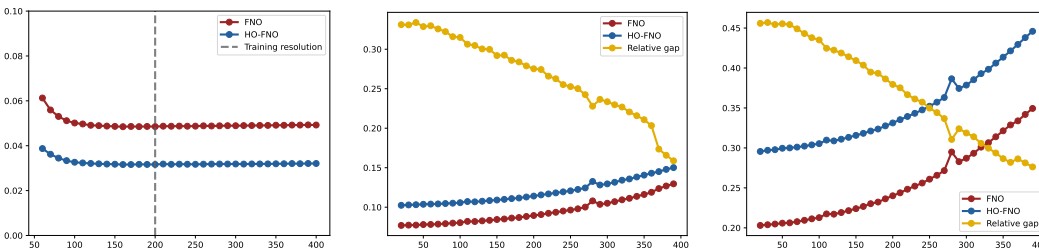

(a) Resolution equivariance of FNO and HO-FNO on the Darcy flow dataset.

(b) Wall-clock time for a single-sample inference at resolution $400 \times 400$.

(c) Wall-clock time for a training batch of size 64, at resolution $400 \times 400$.

Figure 2: Comparison of HO-FNO with FNO. Resolution equivariance is shown on the left (a), where nRMSE in function of the tested resolution. Wall clock time comparisons for inference and training are shown in Figures (b) and (c), where the time is function of the number of retained modes (from 20 to 400). Relative gap is also reported (on a scale from 0 to 1)

From Figure 2, we see that the resolution equivariance property from FNO is kept with our proposed HO-FNO. Furthermore, we observe that mode mixing induces a fixed overhead when compared to FNO. Apart from this fixed additional cost, the time curves for training and inference are the same, with a relative gap decreasing, suggesting our method scales comparably to FNO.

## 6 DISCUSSION AND CONCLUSION

We introduced Higher-Order Fourier Neural Operators (HO-FNO), which augment spectral operator layers with explicit $m$-linear frequency mixing that mimics the polynomial nonlinearities found in many PDEs. Concretely, each retained mode aggregates all $m$-tuples of Fourier coefficients whose indices sum to the index of that mode, yielding an FFT-efficient higher-order spectral convolution that remains in $\mathcal{O}(\texttt{seq\_len}\log(\texttt{seq\_len}))$ complexity per layer, where $\texttt{seq\_len}$ is the input sequence length. This mechanism requires no additional hyperparameters beyond the standard FNO setup and integrates cleanly with existing operator backbones. Empirically, HO-FNO delivers consistent single-step accuracy gains across Burgers, Diffusion–Reaction, and Navier–Stokes, and improves long-horizon rollout stability in most regimes; on spherical data, the analogous HO-SFNO variant also outperforms SFNO.

**Efficiency.** From a computational standpoint, HO-FNO preserves the asymptotic cost of FNO: one forward and one inverse FFT per layer (per channel group), with the $m$-linear interaction effected via pointwise products in physical space. Thus, while there is a small constant-factor overhead from additional pointwise multiplications, the complexity remains $\mathcal{O}(\texttt{seq\_len}\log(\texttt{seq\_len}))$. Architecturally, our models were parameter-matched to baselines and trained under the same budget (100 epochs within $\sim$15 hours on a single A100 for the hardest case), so accuracy gains cannot be ascribed to larger models. In short, HO-FNO trades a modest compute increase for meaningful predictive improvements, without introducing extra tuning knobs.

**Limitations.** First, rollout stability is informative but was not the central optimization target; interpreting rollout scores requires care because normalization and horizon can favor models whose visual fidelity is weak despite low aggregate error. Indeed, UNO and U-Net sometimes report competitive rollout nRMSE while being markedly worse on single-step metrics and visuals, particularly on Diffusion–Reaction.

Second, our formulation is motivated by PDEs with polynomial nonlinearities (quadratic/cubic), for which $m$-linear spectral couplings are a principled inductive bias. Whether similar gains hold for systems dominated by non-polynomial or stiff source terms remains to be established.

Third, we instantiated the linear maps $A_i$ in the higher-order convolution as pointwise operators shared across spatial locations (and not across layers). More expressive choices (e.g., localized kernels, scale-dependent maps, or cross-channel structures) may further improve accuracy but were left for future work.

Finally, although we extended to spherical geometries using generalized Fourier bases, broader validation on irregular meshes or other manifolds would strengthen the case for universality.

**Perspectives** A few natural directions follow.

(i) **Backbone integration:** Combine higher-order spectral mixing with deeper multiscale operators (e.g., UNO-style encoder–decoders) to exploit both cross-scale and cross-mode interactions.

(ii) **Adaptive order and structure:** Learn the effective interaction order $m$ and the parameterization of $A_i$ per layer/task; introduce sparsity or symmetry constraints to reflect known physics.

(iii) **Geometry and physics priors:** Extend to other manifolds/meshes via appropriate spectral bases; couple HO-FNO with conservation or stability regularizers to target rollout fidelity explicitly.

(iv) **Evaluation protocols:** Complement normalized rollout metrics with perceptual/physics-aware scores and standardized horizons to avoid misleading comparisons across models.

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

APPENDIX

## A NOTATIONS

For convenience, we summarize the notation used throughout the paper.

| Symbol | Meaning |
|---|---|
| $\Omega$ | Spatial domain. |
| $d$ | Number of spatial dimensions. |
| $\mathcal{I}$ | Set of $d$-uple of indices $\alpha = (\alpha_1, \ldots, \alpha_d) \in \mathbb{N}^d$, where $\alpha_i$ indicates the order of derivative in the $i$-th dimension, $\frac{\partial^{\alpha_i}}{\partial x_i}$. |
| $\partial\Omega$ | Boundary of the spatial domain. |
| $\mathbb{T}^d$ | $d$-dimensional torus, i.e. periodic domain. |
| $u(x,t) \in \mathbb{R}$ | Solution field at space–time point $(x,t)$ with $C$ channels. |
| $\widehat{u}(k,t) \in \mathbb{C}$ | Fourier coefficient of $u(x,t)$ at frequency $k \in \mathbb{Z}^d$. |
| $i$ | The imaginary number $i = \sqrt{-1}$. |
| $n$ | Degree of nonlinearity ($n = 2$ for quadratic, $n = 3$ for cubic). |
| $N$ | Total n |
| $\nu$ | PDE coefficients (e.g., diffusivity (Burgers), viscosity (Navier-Stokes) or hyperdiffusion coefficient (SWE). |
| $\mathcal{B}(u)$ | Boundary condition operator. |
| $u^0$ | Initial condition. |

## B EXTENDED DERIVATION OF FOURIER MIXING IN NAVIER-STOKES

We present here a detailed discussion of the non-linear interactions on the incompressible Navier-Stokes equation.

The Incompressible Navier-Stokes equation is tipically presented in the following form:

$$\partial_t w(x,t) + u(x,t) \cdot \nabla w(x,t) = \nu \Delta w(x,t) + f(x) \qquad x \in (0,1)^2, \ t \in (0,T] \qquad (22)$$

$$\nabla \cdot u(x,t) = 0 \qquad x \in (0,1)^2, \ t \in [0,T] \qquad (23)$$

$$w(x,0) = w_0(x) \qquad x \in (0,1)^2 \qquad (24)$$

Where $\nabla w(x,t) = \left(\partial_{x_1} w(x,t), \ \partial_{x_2} w(x,t)\right)$ is the gradient of $w$, $\Delta w(x,t) = \partial_{x_1 x_1} w(x,t) + \partial_{x_2 x_2} w(x,t)$ is the Laplacian of $w$ and $\nabla \cdot u = \frac{\partial u_1(x,t)}{\partial x_1} + \frac{\partial u_2(x,t)}{\partial x_2}$ is the divergence of $u$. $u(x,t)$ is the velocity at the point $x$ at time $t$ and $w$ is the vorticity field $w(x,t) = \partial_{x_1} u_2(x,t) - \partial_{x_2} u_1(x,t)$.

**From the velocity to the vorticity formulation** Firstly we will express the PDE in terms of the sole vorticity $w$. To do so we need to express $u$ in function of $w$. By the incompressibility condition $\nabla \cdot u = 0$ implies that exists a function, called streamfunction, $\psi = \psi(x,t)$ such that $u = \nabla^\perp \psi = \left(-\frac{\partial \psi}{\partial x_2}, \frac{\partial \psi}{\partial x_1}\right)$, therefore, by substitution we obtain $w$ in function of the stream function

$$w = \partial_{x_1} u_2 - \partial_{x_2} u_1 = \partial_{x_1}\left(\frac{\partial \psi}{\partial x_1}\right) + \partial_{x_2}\left(\frac{\partial \psi}{\partial x_2}\right) = \Delta \psi \qquad (25)$$

Therefore $\psi$ is obtained from $w$ by solving the Poisson problem $\Delta \psi = w$ in $(0,1)^2$ with appropriate boundary conditions. Once $\psi$ is founded, the velocity $u$ is recovered by $u = \nabla^\perp \psi$ and since $w = \Delta \psi$ we can write $u$ in function of $w$ as $u = \nabla^\perp \Delta^{-1} w$ and same for Navier-Stokes equation:

$$\partial_t(w) = \nu \Delta w(x,t) - (\nabla^\perp \Delta^{-1} w) \cdot \nabla w(x,t) + f(x) \qquad x \in (0,1)^2, \ t \in (0,T] \quad (26)$$

$$\nabla \cdot \nabla^\top \Delta^{-1} w = 0 \qquad\qquad\qquad\qquad\qquad\qquad x \in (0,1)^2, \ t \in [0,T] \quad (27)$$

$$w(x,0) = w_0(x) \qquad\qquad\qquad\qquad\qquad\qquad\qquad x \in (0,1)^2 \quad\quad (28)$$

**Fourier Transform of the Navier-Stokes equation** Now we take the Fourier transform of the vorticity version of the Navier Stokes equation, by taking in consideration that $\widehat{\nabla w}(k,t) = 2\pi i k \cdot \widehat{w}(k,t)$, $\widehat{\Delta w}(k,t) = -(2\pi)^2 |k|^2 \widehat{w}(k,t)$ and $\widehat{w \odot w} = \sum_{q+p=k} \widehat{w}(q,t)\widehat{w}(p,t)$. Therefore equation 26 becomes

$$\partial_t(\widehat{w})(k,t) = -\nu(2\pi)^2 |k|^2 \widehat{w}(k,t) - \sum_{p+q=k} \frac{(p+q) \cdot p^\perp}{|p|^2} \widehat{w}(p,t)\widehat{w}(q,t) + \widehat{f}(k,t) \quad (29)$$

For $k \in \mathbb{Z}^2$, $t \in (0,T]$.

## C DATASETS

| Dataset Name | # Trajectories | # Timesteps | Mesh Type | Resolution |
|---|---|---|---|---|
| Burgers (1D) | 10 000 | 200 | Regular (1D line) | 1024 |
| Diffusion-Reaction (2D) | 1000 | 100 | Regular (2D periodic box) | $128 \times 128$ |
| Navier-Stokes (2D) $\nu = 10^{-4}$ | 10000 | 50 | Regular (2D periodic box) | $64 \times 64$ |
| Navier-Stokes (2D) $\nu = 10^{-5}$ | 1200 | 20 | Regular (2D periodic box) | $64 \times 64$ |
| PlanetSWE (2D) | 50 | 100 | Sphere (latitude-longitude grid) | $256 \times 128$ |

Table 4: Benchmark PDE datasets used in our experiments.

### C.1 1D BURGERS EQUATION

The Burgers' equation is a PDE modeling the non-linear behavior and diffusion process in fluid dynamics as

$$\partial_t u(t,x) + \partial_x \left( \frac{u^2(t,x)}{2} \right) = \frac{\nu}{\pi} \partial_{xx} u(t,x) \qquad\qquad x \in (0,1), \ t \in (0,2] \quad (30)$$

$$u(0,x) = u_0(x) \qquad\qquad\qquad\qquad x \in (0,1) \quad\quad (31)$$

where $\nu$ is the diffusion coefficient, which assumed constant, $\nu = 0.001$ in our dataset. Our dataset use the periodic boundary condition and, as initial condition, we use the following super-position of sinusoidal waves:

$$u_0(x) = \sum_{k_i = k_1, \ldots, k_N} A_i \sin(k_i x + \phi_i) \quad (32)$$

where $k_i = \frac{2\pi n_i}{L_x}$ are wave numbers whose $n_i$ are integer numbers selected randomly in $[1, n_{\max}]$, $N$ is the integer determining how many waves to be added, $L_x$ is the calculation domain size, $A_i$ is a random float number uniformly chosen in $[0,1]$, and $\phi_i$ is the randomly chosen phase in $(0, 2\pi)$.

The numerical solution was calculated with the temporally and spatially 2nd-order upwind difference scheme for the advection term, and the central difference scheme for the diffusion term.

The dataset we considered is provided by PDEBench (Takamoto et al., 2022).

### C.2 2D DIFFUSION-REACTION EQUATION

The 2D diffusion-reaction equation is a PDE modeling two non-linearly coupled variables, namely the activator $u = u(t,x,y)$ and the inhibitor $v = v(t,x,y)$. The activator models a quantity that

promotes or "activates" some process (e.g. chemical concentration in a reaction). The inhibitor models a quantity that suppresses or "inhibits" the process triggered by the activator (e.g. consuming the activator in a chemical reaction). The equation is written as

$$\partial_t u = D_u \partial_{xx} u + D_u \partial_{yy} u + R_u \tag{33}$$
$$\partial_t v = D_v \partial_{xx} v + D_c \partial_{yy} v + R_v \tag{34}$$

where $D_u$ and $D_v$ are the diffusion coefficient for the activator and inhibitor, respectively, $R_u = R_u(u,v)$ and $R_v = R_v(u,v)$ are the activator and inhibitor reaction function, respectively. The domain of the simulation includes $x \in (-1,1)$, $y \in (-1,1)$, $t \in (0,5]$.

The reaction functions for the activator and inhibitor are defined by the Fitzhugh-Nagumo equation (Klaasen & Troy, 1984), written as:

$$R_u(u,v) = u - u^3 - k - v \tag{35}$$
$$R_v(u,v) = u - v \tag{36}$$

where $k = 5 \times 10^{-3}$, and the diffusion coefficients for the activator and inhibitor are $D_u = 1 \times 10^{-3}$ and $D_v = 5 \times 10^{-3}$, respectively. The initial condition is generated as standard normal random noise $u(0,x,y) \sim \mathcal{N}(0,1)$ for $x \in (-1,1)$ and $y \in (-1,1)$.

We employ a no-flow Neumann boundary condition, meaning that

$$D_u \partial_x u = 0 \tag{37}$$
$$D_v \partial_x v = 0 \tag{38}$$
$$D_u \partial_y u = 0 \tag{39}$$
$$D_v \partial_y v = 0 \quad \text{for } x,y \in (-1,1)^2 \tag{40}$$

The spatial discretization is preformed using the finite volume method (LeVeque, 2002), and the time integration is performed using the built-in fourth order Runge-Kutta method in the scipy package (Virtanen et al., 2020).

The dataset on Diffusion-Raction was taken form PDEBench (Takamoto et al., 2022)

### C.3   2D Navier Stokes equations

The 2D Navier-Stokes equation for a viscous, incompressible fluid in vorticity form on the unit torus:

$$\partial_t w(x,t) + u(x,t) \cdot \nabla w(x,t) = \nu \Delta w(x,t) + f(x) \qquad x \in (0,1)^2, \ t \in (0,T] \tag{41}$$
$$\nabla \cdot u(x,t) = 0 \qquad x \in (0,1)^2, \ t \in [0,T] \tag{42}$$
$$w(x,0) = w_0(x) \quad x \in (0,1)^2 \qquad x \in (0,1)^2 \tag{43}$$

The initial condition $w_0(x)$ is generated according to $w_0 \sim \mu$ where

$$\mu = \mathcal{N}(0, 7^{3/2}(-\Delta + 49I)^{-2.5}) \tag{44}$$

with periodic boundary conditions. The forcing is kept fixed:

$$f(x) = 0.1(\sin(2\pi(x_1 + x_2)) + \cos(2\pi(x_1 + x_2))) \tag{45}$$

The equation is solved using the stream-function formulation with a pseudospectral method. First a Poisson equation is solved in Fourier space to find the velocity field. Then the vorticity is differentiated and the non-linear term is computed is physical space after which it is dealiased. Time is advanced with a Crank–Nicolson update where the non-linear term does not enter the implicit part.

All data are generated on a $256 \times 256$ grid and are downsampled to $64 \times 64$. We use a timestep of $10^{-4}$ for the Crank–Nicolson scheme in the data-generated process where we record the solution every $t = 1$ time units.

We use two datasets on Navier-Stokes equations, with viscosity $\nu = 10^{-4}$ and $\nu = 10^{-5}$, provided in (Serrano et al., 2024) (Li et al., 2020).

## C.4 PLANETSWE

The rotated, hyperviscous, forced Shallow Water Equation (SWE) on a sphere is a classical test problem for dynamical systems cores to be used in large-scale weather and climate models as they capture a number of similar phenomena but are better understood and operate at a more practical scale (Williamson et al., 1992). We used the forced hyperviscous equations in two dimensions:

$$\partial_t u(x,t) = -u(x,t) \cdot \nabla_x u(x,t) - g\nabla_x h(x,t) - \nu\nabla_x^4 u(x,t) - 2\Omega \times u(x,t) \tag{46}$$

$$\partial_t h(x,t) = -H\nabla_x \cdot u(x,t) - \nabla_x \cdot (h(x,t)u(x,t)) - \nu\nabla_x^4 h(x,t) + F(x,t) \tag{47}$$

where $\nu$ is the hyper-diffusion coefficient, $\Omega$ is the Coriolis parameter, $u$ is the velocity field, $H$ is the mean height, and $h$ denotes deviation from the mean height. $F$ is a daily/seasonally varying forcing with periods of 24 and 1008 simulation "hour" respectively.

Initial conditions are randomly sampled from ERA5(Hersbach et al., 2020). $u$, $v$, $z$ are taken from the hpa 500 level with $z$ used as $h$ is the shallow water set-up. Prefiltering was performed by executing ten iterations of 50 steps followed by solving a balance BVP. The dataset we used was generated in (McCabe et al., 2023b) and is part of The Well dataset (Ohana et al., 2024a).

The simulations were performed using the spin-weighted spherical harmonic spectral method in Dedalus (Burns et al., 2020) with 500 simulation hours of burn-in where the next three simulation years (3024 hours), were collected for the data set. Integration is performed forward in time using a semi-implicit RK2 integrator. Step-sizes are computed using the CFL-checker in Dedalus. The $3/2$ rule is used for de-aliasing. Background orography is taken from earth orography and passed through mean-pooling three times (until the simulations became stable empirically). Hyperdiffusion is matched at $\ell = 96$.

The original dataset from The Well (Ohana et al., 2024a) contains 120 trajectories of 3024, each consisting of 3024 timesteps at a spatial resolution of $256 \times 512$. For faster training, we restricted our experiments to the first 50 trajectories, truncated to the initial 100 timesteps, and downsampled the spatial resolution to $256 \times 128$ by averaging.

## C.5 DARCY FLOW

Along with the previous time-dependent PDEs we also benchmark against the steady-state of the 2D Darcy Flow equation provided by Li et al. (2020). The equation is the following second-order, linear, elliptic PDE:

$$-\nabla \cdot (a(x)\nabla u(x)) = f(x) \qquad\qquad x \in (0,1)^2 \tag{48}$$

$$u(x) = 0 \qquad\qquad x \in \partial(0,1)^2 \tag{49}$$

with a Dirichlet boundary where $a \in L^\infty((0,1)^2, \mathbb{R}_+)$ is the diffusion coefficient and $f \in L^2((0,1)^2, \mathbb{R})$ is the forcing function. We are interested in learning the operator mapping the diffusion coefficient $a(x)$ to the solution $u(x)$.

# D METRICS DESCRIPTION

We evaluate the predictive performance of our models using the following metrics:

**Mean Squared Error (MSE).** Given ground truth $y \in \mathbb{R}^d$ and prediction $\hat{y} \in \mathbb{R}^d$, the MSE, sometimes called $L_2$-norm, is defined as

$$\text{MSE}(y, \hat{y}) = \frac{1}{d}\sum_{i=1}^{d}\|y_i - \hat{y}_i\|^2. \tag{50}$$

This metric measures the average squared deviation between predictions and targets. It is numerically stable and therefore commonly used as a training loss, as we do in our experiments. At test time, MSE is also informative since it provides a physically meaningful error measure in the original space. However, MSE scales quadratically with multiplicative factors applied to $y$ and $\hat{y}$, and it is affected by the discretization of the domain. As a result, it is not directly comparable across different datasets or resolutions. For this reason, it is often preferred to also report the Normalized Mean Squared Error (NRMSE) at evaluation time.

**Normalized Mean Squared Error (NRMSE).** The RMSE, often called relative $L_2$-norm, is the MSE normalized by the norm of the target:

$$\text{NRMSE}(y, \hat{y}) = \frac{1}{d} \sum_{i=1}^{d} \frac{\|y_i - \hat{y}_i\|^2}{\|y\|^2}. \tag{51}$$

Unlike MSE, which reports squared units, RMSE is expressed in the same units as the target variable. The error magnitude is thus directly comparable to the physical scale of the data, providing a more intuitive sense of accuracy therefore providing a fair comparisons across datasets and resolutions.

**Rollout Error.** Since we deal with time-dependent systems, we evaluate multi-step predictions by iteratively feeding model outputs back as inputs. The rollout error is computed as the average of a choosen loss , $\mathcal{L}$, across all timesteps:

$$\text{Rollout}(y_{1:T}, \hat{y}_{1:T}) = \frac{1}{T} \sum_{t=1}^{T} \mathcal{L}(y_t, \hat{y}_t), \tag{52}$$

where $T$ is the total number of time steps of the dataset. This metric captures error accumulation over long-term forecasts. Even though rollout stability is beyond the scope of this work, it remains informative to assess how new models perform in this setting, which more closely reflects real-world applications than the teacher-forcing setup. For this reason, we report rollout metrics in all our experiments.

# E    ADDITIONAL RESULTS

Table 5: Test performance of FNO and HO-FNO variants (orders up to 3) on Navier–Stokes datasets with and without RMS Norm applied to the multilinear terms. We report the number of parameters, the validation MSE , the normalized MSE ( nRMSE ), the rollout nRMSE as well as the wall-clock time for a single-sample inference and for a training batch of size 64.

| model | | FNO | | HO-FNO | | | DSFNO* | FFNO* | CNO* | GNO* |
|---|---|---|---|---|---|---|---|---|---|---|
| order | | 1 | | 2 | | 3 | | | | |
| RMS Norm | | no | no | yes | no | yes | | | | |
| N. parameters | | 1 085 729 | 1 094 177 | 1 094 177 | 1 098 401 | 1 098 401 | 1.06M | 0.44M | 2.66M | 4.61M |
| **NS** ($\nu = 10^{-3}$) | MSE | $3.0 \times 10^{-7}$ | $2.5 \times 10^{-7}$ | $7.8 \times 10^{-8}$ | $2.1 \times 10^{-7}$ | $\mathbf{7.6 \times 10^{-8}}$ | – | – | – | – |
| | nRMSE | $4.4 \times 10^{-4}$ | $4.0 \times 10^{-4}$ | $\underline{2.8 \times 10^{-4}}$ | $3.8 \times 10^{-4}$ | $\mathbf{2.7 \times 10^{-4}}$ | – | – | – | – |
| | Rollout | $1.2 \times 10^{-2}$ | $1.1 \times 10^{-2}$ | $\underline{1.8 \times 10^{-3}}$ | $9.7 \times 10^{-3}$ | $\mathbf{1.6 \times 10^{-3}}$ | $5.6 \times 10^{-3}$ | $1.1 \times 10^{-2}$ | $2.0 \times 10^{-2}$ | $2.7 \times 10^{-1}$ |
| **NS** ($\nu = 10^{-4}$) | MSE | $2.6 \times 10^{-3}$ | $1.0 \times 10^{-3}$ | $\mathbf{7.9 \times 10^{-4}}$ | $9.8 \times 10^{-4}$ | $\underline{7.9 \times 10^{-4}}$ | – | – | – | – |
| | nRMSE | $2.9 \times 10^{-2}$ | $1.5 \times 10^{-2}$ | $\mathbf{1.3 \times 10^{-2}}$ | $1.5 \times 10^{-2}$ | $\underline{1.3 \times 10^{-2}}$ | – | – | – | – |
| | Rollout | $7.7 \times 10^{-2}$ | $4.8 \times 10^{-2}$ | $\mathbf{4.6 \times 10^{-2}}$ | $4.8 \times 10^{-2}$ | $\underline{4.6 \times 10^{-2}}$ | $6.0 \times 10^{-2}$ | $1.1 \times 10^{-2}$ | $1.1 \times 10^{-1}$ | $5.3 \times 10^{-1}$ |
| **NS** ($\nu = 10^{-5}$) | MSE | $1.8 \times 10^{-2}$ | $\underline{1.7 \times 10^{-2}}$ | $\mathbf{1.7 \times 10^{-2}}$ | $1.8 \times 10^{-2}$ | $1.8 \times 10^{-2}$ | – | – | – | – |
| | nRMSE | $6.7 \times 10^{-2}$ | $\underline{6.5 \times 10^{-2}}$ | $\mathbf{6.5 \times 10^{-2}}$ | $6.8 \times 10^{-2}$ | $6.8 \times 10^{-2}$ | – | – | – | – |
| | Rollout | $1.3 \times 10^{-2}$ | $\underline{1.1 \times 10^{-2}}$ | $\mathbf{1.1 \times 10^{-2}}$ | $1.2 \times 10^{-2}$ | $1.2 \times 10^{-2}$ | – | – | – | – |
| Wall-clock time | inference (ms) | $1.4 \pm 0.12$ | $1.8 \pm 0.14$ | $2.2 \pm 0.22$ | $1.9 \pm 0.19$ | $2.7 \pm 0.31$ | – | – | – | – |
| | training (ms) | $7.91 \pm 0.34$ | $11.2 \pm 0.44$ | $16.1 \pm 0.61$ | $13.4 \pm 0.50$ | $21.9 \pm 0.70$ | – | – | – | – |

\* Original errors reported in Gao et al. (2025), single-step metrics and results for Navier–Stokes with $\nu = 10^{-5}$ were not provided.

# F    ROLLOUT VISUALIZATIONS

In this section, we present visualizations of the rollout predictions corresponding to Table 1. Across all datasets, HO-FNO consistently produces visually superior results. For the Diffusion–Reaction PDEs (Figures 5 and 7), none of the models accurately capture the dynamics from time 0 to 100. Nevertheless, HO-FNO is able to recover the high-level structure of the solution. For this dataset, to enable a fairer comparison, we additionally report rollout visualizations from time 25 to 100 (Figures 6 and 8), where all models achieve more accurate predictions, thus providing a clearer benchmark for visual assessment.

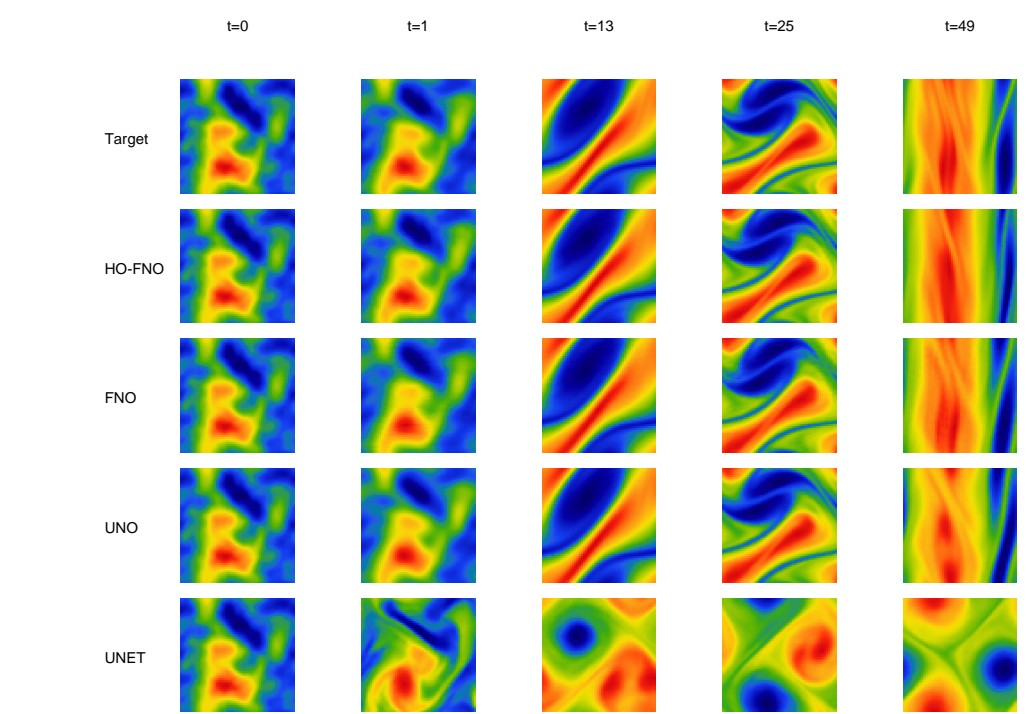

Figure 3: Visualization of Rollout predictions on Navier Stokes with $\nu = 10^{-4}$.

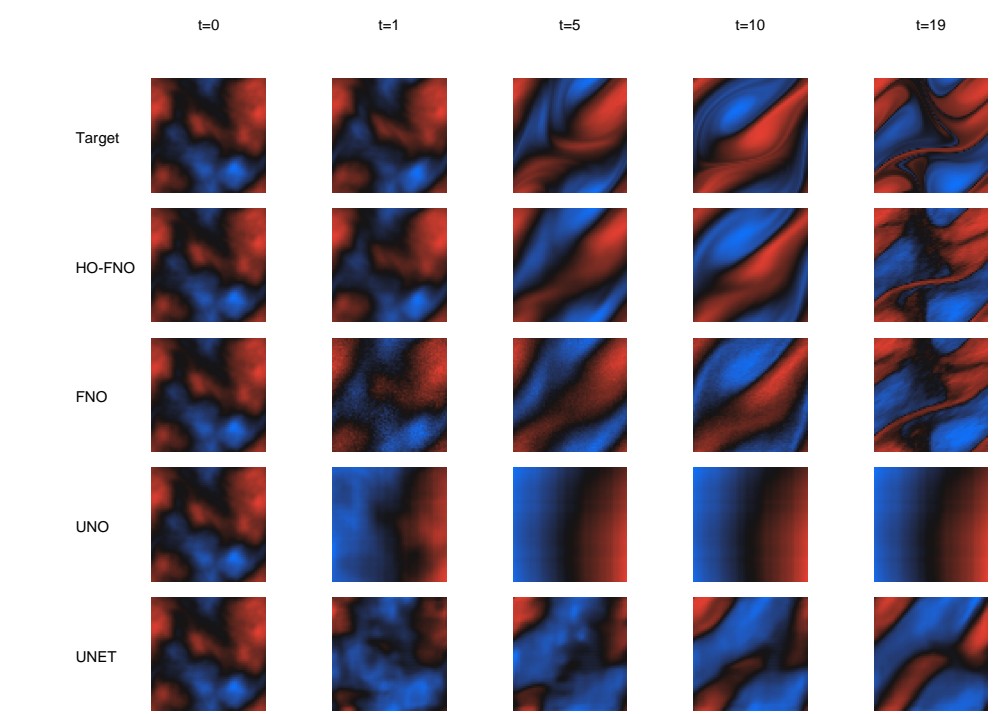

Figure 4: Visualization of Rollout predictions on Navier Stokes with $\nu = 10^{-5}$.

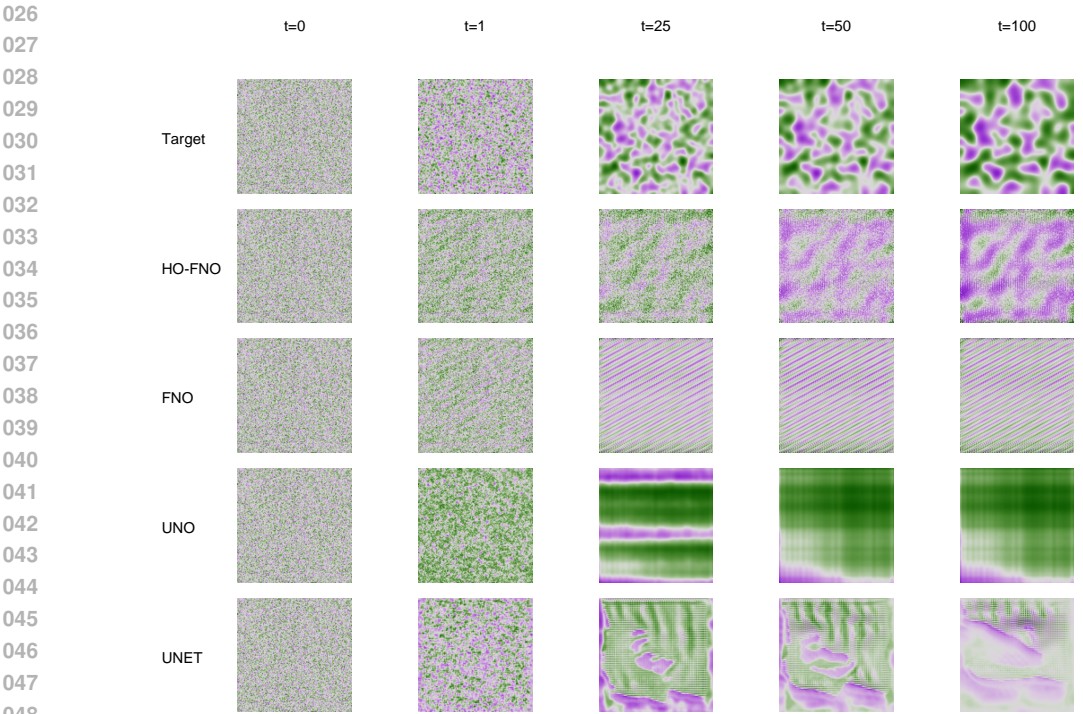

Figure 5: Visualization of Rollout predictions of the activator in the Diffusion-Reaction equation.

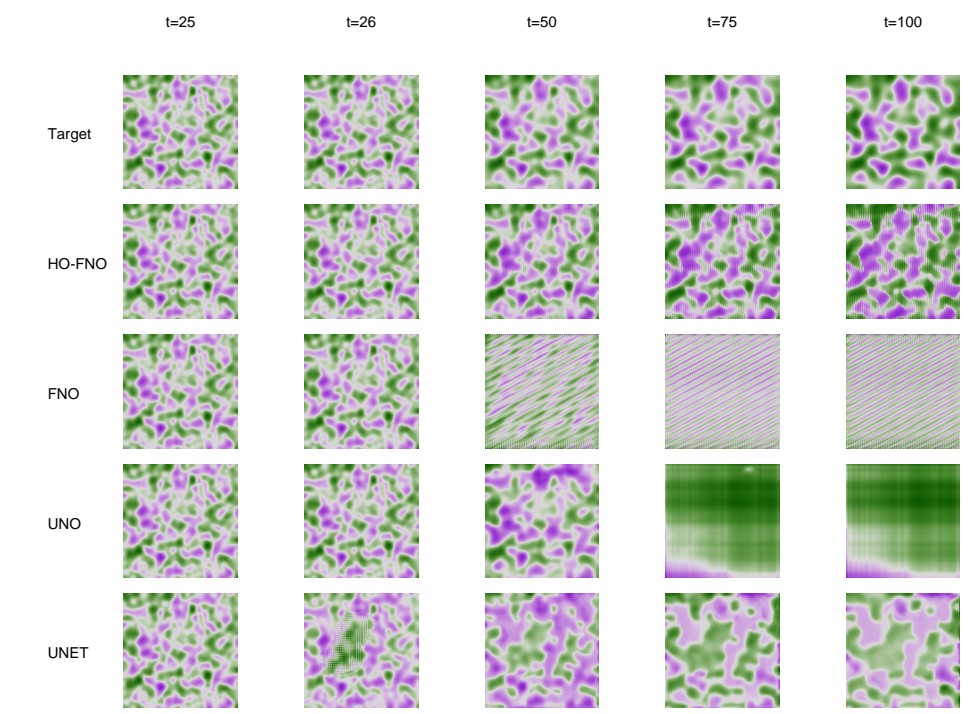

Figure 6: Visualization of Rollout predictions of the activator in the Diffusion-Reaction equation with rollout starting at time 25.

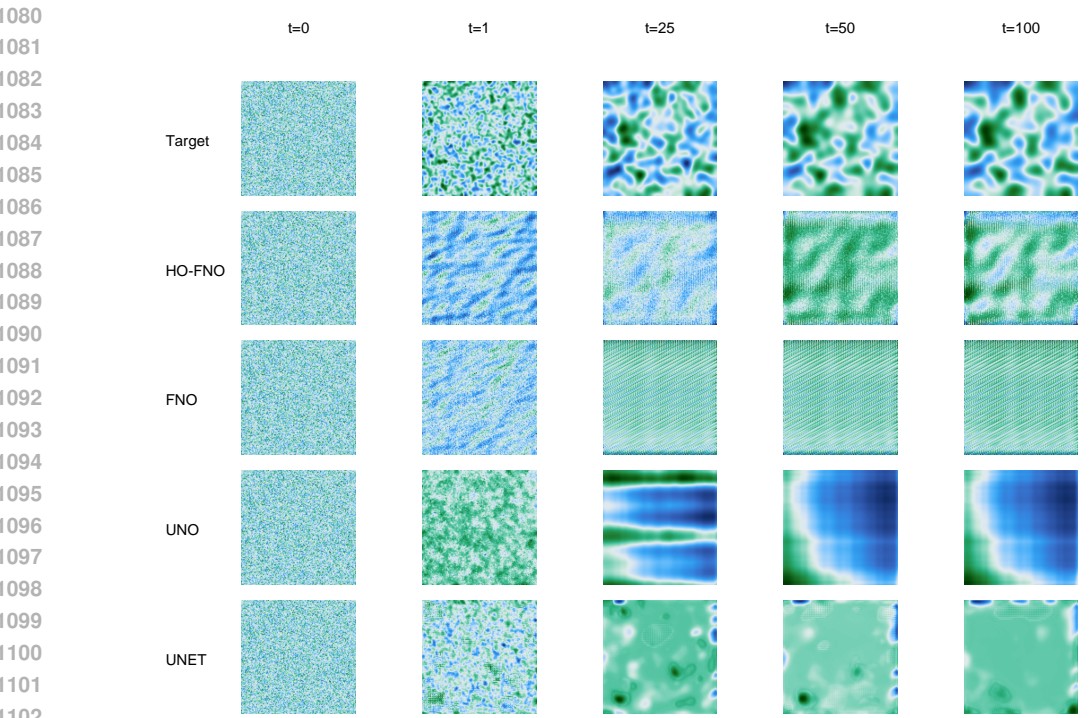

Figure 7: Visualization of Rollout predictions of the inhibitor in the Diffusion-Reaction equation.

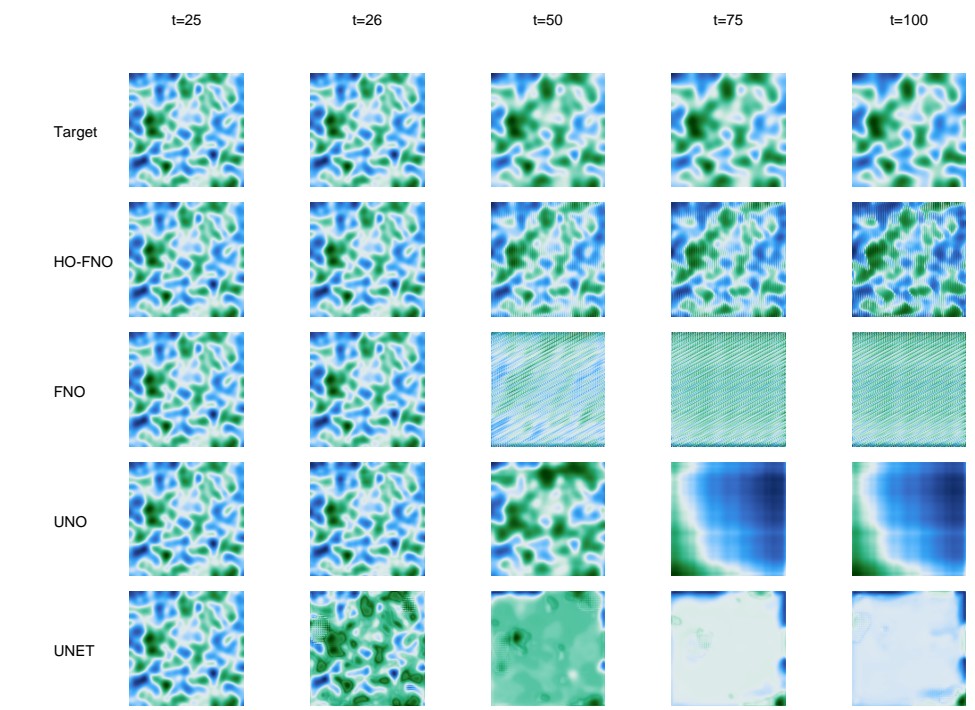

Figure 8: Visualization of Rollout predictions of the inhibitor in the Diffusion-Reaction equation with rollout starting at time 25.

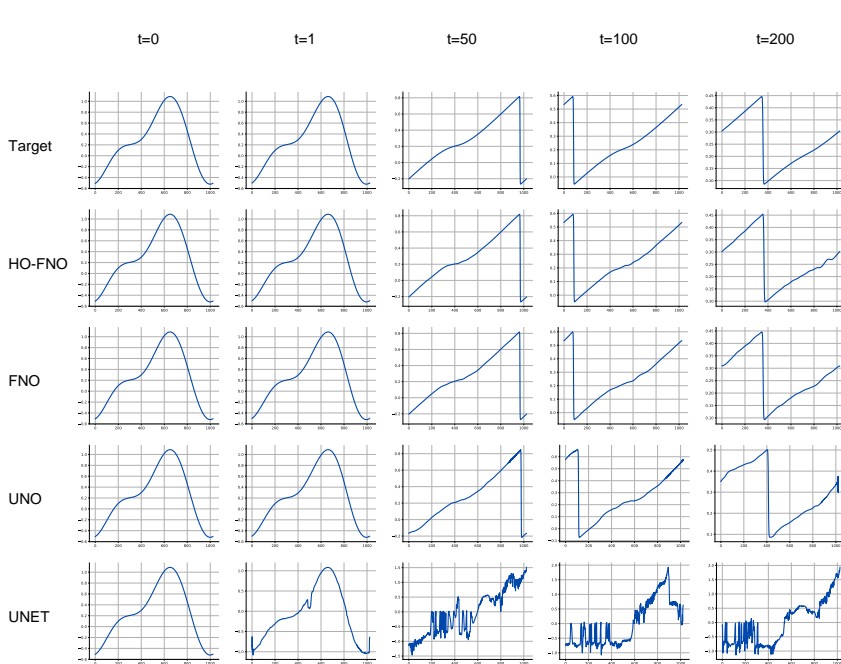

Figure 9: Visualization of Rollout predictions of the inhibitor in the Burgers equation with rollout.

