# OpenReview forum: "Higher-Order Fourier Neural Operator: Explicit Mode Mixer for Nonlinear PDEs"
_ICLR.cc/2026/Conference — Submitted to ICLR 2026_

### Official Review · Reviewer_g6mf · 2025-10-26

**Soundness:** 2
**Presentation:** 2
**Contribution:** 2
**Rating:** 2
**Confidence:** 5

**Summary:**

This paper introduces Higher-Order Fourier Neural Operators (HO-FNO), an extension of the popular Fourier Neural Operator that explicitly models nonlinear mode interactions in the spectral domain. The key insight is elegant: while standard FNO processes Fourier modes independently (suitable for linear PDEs), nonlinear PDEs exhibit structured interactions between modes through polynomial nonlinearities. The authors address this by introducing an m-linear spectral convolution that aggregates m-tuples of Fourier coefficients whose indices sum to each target mode, directly mirroring the triadic interactions seen in equations like Navier-Stokes. The method maintains FFT efficiency at O(N log N) complexity while achieving consistent accuracy improvements across multiple benchmarks including Burgers, Navier-Stokes, Diffusion-Reaction, and spherical Shallow Water equations.

**Strengths:**

I think the authors are trying to address a real limitation of standard FNO, and I appreciate that they're thinking carefully about the physical structure of PDEs rather than just throwing more parameters at the problem. The observation that nonlinear PDEs have structured frequency interactions is valid, and building this into the architecture as an inductive bias is reasonable in principle. The mathematical framework connecting polynomial nonlinearities to m-linear Fourier mixing is clearly presented in Sections 2-3, particularly the detailed Navier-Stokes derivation showing triadic interactions, even if I'm not fully convinced it provides practical benefits beyond what existing methods achieve. I also think it's good that the authors tested on multiple benchmarks rather than cherry-picking a single favorable result the diversity of PDEs (1D Burgers, 2D Navier-Stokes at different viscosities, reaction-diffusion systems, and spherical equations) does give some confidence that the approach has breadth, and the effort to extend the method to non-Euclidean geometries via generalized Fourier transforms shows the authors are thinking about broader applicability.

**Weaknesses:**

The paper has fundamental issues with both theoretical justification and experimental rigor. Theoretically, the contribution is unclea the authors motivate HO-FNO by pointing to triadic interactions in PDEs, but they never rigorously prove what additional expressivity this provides beyond standard FNO with pointwise nonlinearities, which already has universal approximation capabilities. The claim of "first spectral neural operators modeling exact mode interaction" feels like an overstatement when physics-informed methods already encode PDE structure directly. The experimental setup raises serious concerns: there's no statistical significance testing, no error bars reported. The "parameter-matched" claim is never substantiated with actual parameter counts or FLOPs comparisons. The computational complexity analysis is misleading—while FFT operations are O(N log N), the m-linear mixing in physical space adds overhead that isn't carefully quantified. Real wall-clock time comparisons are absent. The results themselves are inconsistent and sometimes contradictory: UNO achieves much better rollout on Diffusion-Reaction (1.59 vs 2.37) despite weaker single-step metrics, which the authors hand-wave away rather than properly investigate.

Moreover, the scope and validation are limited. The method only works for polynomial nonlinearities, excluding many important PDEs with exponential, logarithmic, or other non-polynomial terms. There are no ablation studies systematically varying m, the number of modes M, or the parameterization of the A_i operators. The claim that hyperparameters don't need tuning is convenient but completely unsupported. No comparison to simply making FNO deeper with the same parameter budget, and no comparison to higher-order attention mechanisms despite citing them as related work. The improvements shown are often marginal and may not be practically meaningful. The spherical harmonics extension (HO-SFNO) is straightforward and expected, adding little conceptual depth. The presentation also suffers from inconsistent notation, missing experimental details (initialization, learning rates, batch sizes), and visualizations that are too small to meaningfully evaluate. Overall, this feels like an early-stage idea that needs substantial additional work before it's ready for publication.

**Questions:**

1. **Theoretical Expressivity vs. Standard FNO**: Can you provide some theoretical analysis demonstrating what functions or PDE solutions HO-FNO can represent that standard FNO with sufficient depth cannot (I would assume they are the same though)? Given that both architectures have universal approximation capabilities, what is the formal advantage of explicit m-linear mixing versus implicit mixing through layered pointwise nonlinearities? Without this clarity, it's difficult to assess whether the improvements are fundamental or simply due to a different parameterization.

2. **Experimental Rigor and Fair Comparison**: Your claim of "parameter-matched" models is central to demonstrating that improvements come from the architectural innovation rather than simply having more capacity. Can you provide: (a) exact parameter counts for each model on each dataset, (b) actual wall-clock training and inference times, (c) results across multiple random seeds with error bars and statistical significance tests, and (d) a comparison against simply making FNO deeper/wider with the same parameter budget? The current experimental setup makes it impossible to determine whether HO-FNO's gains are meaningful or within noise margins.

3. **Inconsistent Rollout Performance and Metric Interpretation**: In Table 1, UNO achieves substantially better rollout performance on Diffusion-Reaction (1.59 vs your 2.37) despite much worse single-step metrics, yet your visual analysis suggests UNO's predictions are qualitatively poor. This contradiction raises concerns about what these metrics actually measure. Can you explain: (a) why rollout NRMSE and visual quality sometimes disagree so dramatically, (b) whether the normalization scheme in your rollout metric might be masking important error patterns, and (c) how practitioners should interpret these conflicting signals when choosing a model? This seems like a fundamental issue with the evaluation protocol that needs resolution.

---

> ### Author Response · Authors · 2025-11-24
> **Part 1.**
>
> We would like to thank the reviewer for taking the time to review our work and for the comments that were helpful to improve our work.
>
> > Weakness 1 & Question 1.
>
> We agree with the reviewer that FNO is already a universal approximator and that a theoretical argument on the efficacy of our additions would be interesting. Since FNO is a universal approximator, our method is a universal approximator as well, since it is a strict extension of the former (an appropriate choice of our additional parameters reduces exactly to FNO for any given choice of the shared weights). Consequently, no stronger “density” or universality theorem can be obtained.  A task to showcase the benefit of our method could consist of finding a certain class of tasks (such as PDEs of nonlinear order 2 or a more manageable subset of these) and show that to obtain a fixed (train or test) error of epsilon requires a substantially higher number of layers (as a function of epsilon) in a standard FNO than in a higher-order FNO. However, we were not able to get such a theoretical result. Some of the main difficulties are given by the pointwise non-linearities that already induce a global, unstructured, and relatively weak form of mode mixing, making it difficult to formalize a clean separation and avoiding restrictive assumptions that would make the result irrelevant for realistic settings such as the one of our experiments.
>
> > Weakness 2.
>
> We agree on toning down the claim "first spectral neural operators modeling exact mode interaction". However we are not aware of other works that embed the non-linear order of the PDE as a structural prior in the architectural design. Physics-informed methods (i.e. PINNs and PINOs) do not encode informations on the PDEs in the design of the model, instead they add the residual of the equation to the loss as a regularization term. This allows to train models with few or no data (instead of in a fully data-driven setting as in our case) but it’s, in general, not related to the architecture as our proposed method is. Additionally, our HO-FNO can be trained in a physics-informed style with no modifications of the architecture. Please advise us if we misunderstood your comment.
>
> > Question 2 (a).
>
> We added at the end of section 3 (page 4) a parameter count section where we observe that the number of parameters in a spectral convolution of FNO is $MC^2$  where $C$ denotes the number of input and output channels, and $M$ the total number of retained Fourier modes in each spatial dimension. Our HO-FNO of order $m$ introduces $m$ additional matrices of size $C \times C$ applied pointwise in space. Therefore the total parameter count of a HO-Spectral convolution is $MC^2 + mC^2$ that is negligible as confirmed by the exact parameter count we added in the new Table 2 due to the fact than the order of interaction $m$ is typically much smaller than the number of retained Fourier coefficients $M$. We also found that to normalize the multilinear terms improves training stability and performance therefore we added results with and without RMS normalization to the new Table 2.
>
> **Table 2: Test performance of FNO and HO-FNO variants (orders up to $3$) on Navier--Stokes datasets with and without RMS Norm applied to the multilinear terms.**
>
> | Model / Metric | FNO | HO-FNO (2, no RMS) | HO-FNO (2, yes RMS) | HO-FNO (3, no RMS) | HO-FNO (3, yes RMS) | DSFNO* |
> |---|---:|---:|---:|---:|---:|---:|
> | **Parameters** | 1,085,729 | 1,094,177 | 1,094,177 | 1,098,401 | 1,098,401 | 1.06M |
> | **NS (ν = 10⁻³)** MSE | 3.0e-7 | 2.5e-7 | 7.8e-8 | 2.1e-7 | **7.6e-8** | -- |
> | **NS (ν = 10⁻³)** nRMSE | 4.4e-4 | 4.0e-4 | 2.8e-4 | 3.8e-4 | **2.7e-4** | -- |
> | **NS (ν = 10⁻³)** Rollout | 1.2e-2 | 1.1e-2 | 1.8e-3 | 9.7e-3 | **1.6e-3** | 5.6e-3 |
> | **NS (ν = 10⁻⁴)** MSE | 2.6e-3 | 1.0e-3 | **7.9e-4** | 9.8e-4 | 7.9e-4 | -- |
> | **NS (ν = 10⁻⁴)** nRMSE | 2.9e-2 | 1.5e-2 | **1.3e-2** | 1.5e-2 | 1.3e-2 | -- |
> | **NS (ν = 10⁻⁴)** Rollout | 7.7e-2 | 4.8e-2 | **4.6e-2** | 4.8e-2 | 4.6e-2 | 6.0e-2 |
> | **NS (ν = 10⁻⁵)** MSE | 1.8e-2 | 1.7e-2 | **1.7e-2** | 1.8e-2 | 1.8e-2 | -- |
> | **NS (ν = 10⁻⁵)** nRMSE | 6.7e-2 | 6.5e-2 | **6.5e-2** | 6.8e-2 | 6.8e-2 | -- |
> | **NS (ν = 10⁻⁵)** Rollout | 1.3e-2 | 1.1e-2 | **1.1e-2** | 1.2e-2 | 1.2e-2 | -- |
> | **Inference time (ms)** | 1.4 ± 0.12 | 1.8 ± 0.14 | 2.2 ± 0.22 | 1.9 ± 0.19 | 2.7 ± 0.31 | -- |
> | **Training time (ms)** | 7.91 ± 0.34 | 11.2 ± 0.44 | 16.1 ± 0.61 | 13.4 ± 0.50 | 21.9 ± 0.70 | -- |
>
> *DSFNO values from the original paper; single-step metrics and ν = 10⁻⁵ results were not provided.

---

> > ### Author Response · Authors · 2025-11-24
> > **Part 2.**
> >
> > > Question 2 (b).
> >
> > We added a new Table 2 downsizing our models to ~1M parameters for the Navier-Stokes datasets (that now increased to $3$) to enable a fair comparison with the original FNO paper and existing recent literature that is relevant to our work such as DSFNO. We present there also the performances of our version of order 3 and we report in Appendix E an extended version of the same table with additional baselines. For all the models we report wall-clock time for a single-sample inference and for training of a batch of 64 samples. For wall-clock time we report mean and standard deviation averaged over $100$ runs conducted on a GPU Nvidia A100.
> >
> > > Question 2 (c).
> >
> > Due to computational constraints, we were not able to ablate on different hyperparameters (width, depth, number of retained Fourier coefficients) for a given parameter count.  However the experiments in table 1 were conducted following the setup of relevant previous works (FNO, DSFNO). Notably, all models in Table 1 were trained for 500 epochs (as opposed to 100 in the Table 1, already present in our work before this revision) and longer training widened the performance gap indicating that the improvement is unlikely to stem from noise or undertraining. We note that our FNO baseline performs better than what was reported in the original FNO paper, indicating that our comparisons are made under fair baseline settings. The improved performance of our reimplementation of FNO are due to a more modern design and optimization pipeline. Specifically, we adopt GeLU (instead of ReLU) as activation function, and we optimize with AdamW (instead of Adam).
> >
> > We also agree that to report the standard deviation is important for a statistically grounded assessment, however, since each one of our experiments exceeds $10$ hours of training on a GPU Nvidia A100 we do not have them for all the experimental settings we have considered, but we are currently running them and will add them as soon as possible..
> >
> > > Question 3.
> >
> > We agree that rollout metrics can be, sometimes, counterintuitive. Our work focuses on designing an architecture that can better capture the structure of the operator we aim to learn, it is therefore a work on approximation power and single-step metrics can better capture the improvement brought by our proposed method. We do report performances in the rollout regime since it is a critical metric in the field, as in practice, when the model is deployed, only the initial conditions are provided, and the full trajectory needs to be predicted without additional supervision. However, single-step performances do not transfer directly to rollout ones. In other words, two models with the same single-step metrics can have very different rollout performances. Naively if one of the two models performs worse than the other in the first timesteps of the simulation, it will have also a worse rollout performance since errors accumulate. More importantly, even if two models have same single-step performance per time, their rollout can differ drastically. This is mainly due to the issue of model stability, similarly to the stability of numerical methods or to adversarial robustness in other areas of ML such as image classification. Since a stable rollout is not the primary objective of our contribution we do not adopt any trick to improve the rollout metrics (such as Gaussian smoothing, adversarial training or the push-forward method) and we train optimizing just on single-step MSE to present a comparison with the baselines in a clean setting. Also, we didn’t design HO-FNO for stable rollout, however we think it is relevant to show that HO-FNO performs better than the baselines also when unrolled, so for this reason we added the rollout metric.
> >
> > We highlight that also in other works similar counterintuitive behaviors are reported. An example is Table 1 in (mccabe et al., 2025). There, for the Gray-Scott dataset, Poseidon-L has a single step VRMSE that is $1.27$ times smaller than DPOT-H while having a rollout VRMSE from time 1 to time 20 that is $3.29$ times bigger and a rollout loss between time 21 and 60 that is $5.07$ bigger. This phenomenon can be observed many other times in that table and between all the 4 models benchmarked. Therefore this phenomenon is not uncommon and certainly needs to be better understood by the community but it is beyond the scope of this contribution.
> >
> > Also, as you pointed out, the choice of the underlying metric is crucial. We personally believe that MSE ($L^2$) provides numbers more aligned with the visual assessment, however NRMSE (Normalized L^2) is more common as a metric since it enables a more fair comparison between datasets and discretizations.

---

> > > ### Author Response · Authors · 2025-11-24
> > > **Part 3.**
> > >
> > > > Other Weaknesses.
> > >
> > > As far as we know, attentions of order bigger than $2$ were never introduced in a model dedicated to physical simulation. Additionally, an attention of order 3 has cubic complexity in the number of mesh points making it unfeasible to apply it naively. We could take inspiration from the literature on reducing the computaitonal cost of transformers to apply higher order attentions to our setting at a feasible cost, but to the best of our knowledge it has never been done and it would be a different contribution in itself. Therefore we do not provide such experimental comparisons.
> > >
> > > We agree that the extension to spherical harmonics is not technically complicated and the method is expected to retain its advantages when operating on a generalized Fourier basis. However, we consider insightful to demonstrate it empirically.
> > >
> > >  > Additional updates.
> > >
> > > Additionally, we have improved the quality of Figure 1 and empirically validated resolution-equivariance by training on Darcy Flow at resolution $200 \times 200$ and testing at resolutions between $50 \times 50$ and $400 \times 400$.
> > >
> > > > References.
> > > ```
> > > @misc{mccabe2025walruscrossdomainfoundationmodel,
> > > title={Walrus: A Cross-Domain Foundation Model for Continuum Dynamics},
> > > author={Michael McCabe and Payel Mukhopadhyay and Tanya Marwah and Bruno Regaldo-Saint Blancard and Francois Rozet and Cristiana Diaconu and Lucas Meyer and Kaze W. K. Wong and Hadi Sotoudeh and Alberto Bietti and Irina Espejo and Rio Fear and Siavash Golkar and Tom Hehir and Keiya Hirashima and Geraud Krawezik and Francois Lanusse and Rudy Morel and Ruben Ohana and Liam Parker and Mariel Pettee and Jeff Shen and Kyunghyun Cho and Miles Cranmer and Shirley Ho}, year={2025}, eprint={2511.15684}, archivePrefix={arXiv},  primaryClass={cs.LG},  url={https://arxiv.org/abs/2511.15684}}
> > > ```

---

> > > > ### Comment · Reviewer_g6mf · 2025-11-28
> > > > **Response to Authors**
> > > >
> > > > Thanks to the authors for their detailed response and substantial revisions. The new parameter-count discussion and Table 2 make the “parameter-matched” claim much clearer, and the added wall-clock timings on A100 GPUs address my concerns about hidden computational costs. The ablations over interaction order and RMSNorm help clarify where the gains are coming from, and the expanded discussion of rollout vs single-step metrics (with qualitative examples and a connection to stability) makes the evaluation story more coherent. The framing has also shifted toward “architectural inductive bias for polynomial nonlinearities,” with limitations better acknowledged and the spherical extension presented more modestly.
> > > >
> > > > That said, some of my main concerns are only partially resolved. I understand that proving a clean expressivity or sample-efficiency separation from standard FNO is technically hard in this setting, and I appreciate the authors’ honesty about this. However, the other issues remain: the experiments are still single seed with no error bars, and there is no direct comparison to a deeper/wider FNO under the same parameter budget, which makes it hard to assess robustness and whether the gains are truly structural. The method also remains tailored to polynomial nonlinearities, with no evidence for more general PDE classes. Overall, I view this as a promising architectural idea with improved but still not fully convincing evidence. In light of the revisions, I am raising my recommendation from reject to weak reject.

---

### Official Review · Reviewer_tTX4 · 2025-10-28

**Soundness:** 2
**Presentation:** 2
**Contribution:** 1
**Rating:** 2
**Confidence:** 5

**Summary:**

Summary.
This paper proposes Higher-Order Fourier Neural Operators (HO-FNO), which extend FNOs by introducing an $m$-linear spectral convolution that explicitly mixes Fourier modes reflecting the polynomial nonlinearities in PDEs. The method retains FFT-level $O(N\log N)$ complexity, generalizes to spherical harmonics (HO-SFNO), and achieves improved single-step accuracy and stable rollouts on Burgers, Diffusion--Reaction, Navier--Stokes, and spherical SWE benchmarks.

Contributions.
A higher‑order spectral convolution that performs explicit $m$‑linear mode mixing inside neural‑operator layers; an FFT‑efficient implementation that preserves FNO‑class complexity without adding hyperparameters; and a geometry‑aware extension to spherical harmonics (HO‑SFNO) validated by empirical improvements across standard nonlinear PDE benchmarks.

**Strengths:**

1) Originality: Physics-aligned higher-order spectral convolution explicitly mixing modes with $k_1+\cdots+k_m=k$, introducing a new operator family beyond per-mode scaling.

2) Quality: Efficient layer design using one FFT and one IFFT with $\mathcal{O}(N\log N)$ complexity and no extra hyperparameters, delivering strong accuracy with comparable model size.

3) Clarity: Equations connect the physical $m$-linear product to Fourier-space mixing $k_1+\cdots+k_m=k$; the layer is simple to implement and its assumptions are clearly stated.

4) Significance: Gains on nonlinear flows and a spherical variant (HO--SFNO) that outperforms SFNO on $S^2$, indicating impact for operator learning and geophysical modeling.

**Weaknesses:**

1) Abstract over-claim. The abstract asserts that linear PDEs have Fourier modes that “evolve independently.” This only holds in special cases (e.g., constant coefficients on periodic domains where the operator is diagonal in the Fourier basis). In general (variable coefficients, non-periodic geometries, alternative bases), linear PDEs can couple modes; the claim should be qualified.

2) Resolution-equivariance phrasing. The introduction attributes “resolution-equivariant” behavior to neural operators broadly; in practice, this property is specific to FNO/SNO-style constructions under appropriate truncation/aliasing assumptions, not to all neural-operator families. Please narrow or define the scope precisely.

3) Novelty vs.\ prior TMLR work (DSFNO). The claimed contribution overlaps with prior work that already introduces non-diagonal frequency-domain interactions and formalizes mode mixing. The contribution should be positioned as a structured, higher-order ($m$-linear) mixer aligned with polynomial PDE couplings, with direct comparisons and discussion.

4) Parameters and fairness. Each HO-FNO layer adds $m$ per-point channel maps $A_i\in\mathbb{R}^{C\times C}$ plus pointwise products; at fixed depth/width this likely increases parameters relative to a standard FNO layer. Please provide explicit per-layer and total parameter counts (FNO vs.\ HO-FNO/HO-SFNO) and a table confirming parameter matching in the reported experiments.

5) Ablations are missing. The paper does not isolate (i) the interaction order $m$ (e.g., $m\!=\!1,2,3$), (ii) the role/number of $A_i$, or (iii) the number of retained modes $K$. Without these ablations, it is difficult to attribute gains specifically to higher-order mixing (vs.\ capacity).

6) Clarity and notation. The concurrent use of “$n$-linear” (PDE) and “$m$-linear” (model) can be confusing; add a brief notation paragraph. In Figure~1, denote products with $\times$ (instead of asterisks) and display the constraint $k_1+\cdots+k_m=k$ near the mixer to make the interaction explicit and improve figure quality.

References.
Gao, W., Luo, J., Xu, R., and Liu, Y. Dynamic Schwartz–Fourier Neural Operator for Enhanced Expressive Power. Transactions on Machine Learning Research (TMLR), 2025.

**Questions:**

1) Will you qualify the abstract’s “independent modes for linear PDEs” statement and specify the conditions under which operators are diagonal in Fourier space?

2) Can you provide a precise contrast to DSFNO: interaction form, symmetry implications, computational complexity, and regimes where each approach is preferable?

3) Please add ablations on $m$, on removing/tying $A_i$, and on the number of retained modes $K$ (and, if feasible, constant vs.\ learned spectral multipliers).

4) Please report closed-form parameter counts per layer (in $C$, $m$, groups, $K$) and a summary table verifying parameter matching across baselines.

5) Minor presentation: in Figure~1, switch to $\times$ for products and print the $k_1+\cdots+k_m=k$ constraint on the diagram for readability.

References.
Gao, W., Luo, J., Xu, R., and Liu, Y. Dynamic Schwartz–Fourier Neural Operator for Enhanced Expressive Power. Transactions on Machine Learning Research (TMLR), 2025.

---

> ### Author Response · Authors · 2025-11-24
> **Part 1.**
>
> We would like to thank the reviewer for taking the time to review our work and for the comments that were helpful to improve our work.
>
> > Weakness 1 & Question 1.
>
> We thank the reviewer for pointing this out and agree that the statement in the abstract is wrong as written. Our intention was to refer to the classical setting of constant-coefficient linear PDEs on periodic domains (as in the standard FNO benchmarks), where the corresponding operator is diagonal in the Fourier basis and each mode indeed evolves independently. We corrected this statement in the revised version of the submission.
>
> > Weakness 2.
>
> In our work we adopt the definition of Neural operators given in (Kovachki et al., 2023) and more recently restated in (Berner et al., 2025), where a Neural operator, to be called as such, should follow the following 3 properties: Discretization-agnosticity, Fixed number of parameters, Universal approximation. If these properties are not satisfied, we follow these two works to say that the model is not learning and predicting the underlying solution operator but just one of its discretization at a fixed resolution and should therefore just be called a neural network. In (Berner et al., 2025) Graph neural operators and Transformer neural operators (such as OFormer) are included in the class of Neural Operators. However, we agree with the reviewer that graph- and attention-based models typically underperform FNO-like variants in zero-shot super resolution. To improve the precision of our work we modified the paragraph in the introduction by stating that neural operators show approximate resolution-equivariance and we mentioned, in the successive paragraph, that, among neural operators, FNO stands out for its ability to transfer across resolutions. We acknowledge that FNO is not aliasing-free but we did not emphasize this aspect in the submission since it is not the main focus of our work. We are open to develop more on this aspect if considered of interest.
>
> > Weakness 3 & Question 2.
>
> We agree that DSFNO is very relevant to our work, at the time of writing we were not aware of this paper since it was published in TMLR in June 2025. However, the method differs from our proposal at a fundamental level. DSFNO’s convolution has a kernel that, differently from the static one of standard FNO, depends on the input via a hyper-network that maps the truncated Fourier coefficients of the inputs to the kernel applied via a classic spectral convolution. Therefore, the kernel used by DSFNO is unstructured and agnostic with respect to the order of non-linearity of the PDE. In contrast, our method is strongly structured, and needs very few additional parameters e.g., 0.07% more for HO-FNO against the 14% more of DSFNO as specified in their paper. Note that we added the exact parameter count for our models to the new table 2. Additionally we added Table 2 with results on the three Navier-Stokes datasets (we added NS with viscosity $\nu = 10^{-3}$) with the same setting of DSFNO and FNO. Our best HO-FNO model performs $2.4$ times better than DSFNO on the common benchmarks and a wider gap is also observed with all the other baselines we report in table 2. We have also added an extended table with more baselines in appendix E. We also found that to normalize the multilinear terms improves training stability and performance therefore we added results with and without RMS normalization to the new Table 2.
>
> **Table 2: Test performance of FNO and HO-FNO variants (orders up to $3$) on Navier--Stokes datasets with and without RMS Norm applied to the multilinear terms.**
>
> | Model / Metric | FNO | HO-FNO (2, no RMS) | HO-FNO (2, yes RMS) | HO-FNO (3, no RMS) | HO-FNO (3, yes RMS) | DSFNO* |
> |---|---:|---:|---:|---:|---:|---:|
> | **Parameters** | 1,085,729 | 1,094,177 | 1,094,177 | 1,098,401 | 1,098,401 | 1.06M |
> | **NS (ν = 10⁻³)** MSE | 3.0e-7 | 2.5e-7 | 7.8e-8 | 2.1e-7 | **7.6e-8** | -- |
> | **NS (ν = 10⁻³)** nRMSE | 4.4e-4 | 4.0e-4 | 2.8e-4 | 3.8e-4 | **2.7e-4** | -- |
> | **NS (ν = 10⁻³)** Rollout | 1.2e-2 | 1.1e-2 | 1.8e-3 | 9.7e-3 | **1.6e-3** | 5.6e-3 |
> | **NS (ν = 10⁻⁴)** MSE | 2.6e-3 | 1.0e-3 | **7.9e-4** | 9.8e-4 | 7.9e-4 | -- |
> | **NS (ν = 10⁻⁴)** nRMSE | 2.9e-2 | 1.5e-2 | **1.3e-2** | 1.5e-2 | 1.3e-2 | -- |
> | **NS (ν = 10⁻⁴)** Rollout | 7.7e-2 | 4.8e-2 | **4.6e-2** | 4.8e-2 | 4.6e-2 | 6.0e-2 |
> | **NS (ν = 10⁻⁵)** MSE | 1.8e-2 | 1.7e-2 | **1.7e-2** | 1.8e-2 | 1.8e-2 | -- |
> | **NS (ν = 10⁻⁵)** nRMSE | 6.7e-2 | 6.5e-2 | **6.5e-2** | 6.8e-2 | 6.8e-2 | -- |
> | **NS (ν = 10⁻⁵)** Rollout | 1.3e-2 | 1.1e-2 | **1.1e-2** | 1.2e-2 | 1.2e-2 | -- |
> | **Inference time (ms)** | 1.4 ± 0.12 | 1.8 ± 0.14 | 2.2 ± 0.22 | 1.9 ± 0.19 | 2.7 ± 0.31 | -- |
> | **Training time (ms)** | 7.91 ± 0.34 | 11.2 ± 0.44 | 16.1 ± 0.61 | 13.4 ± 0.50 | 21.9 ± 0.70 | -- |
>
> *DSFNO values from the original paper; single-step metrics and ν = 10⁻⁵ results were not provided.

---

> > ### Author Response · Authors · 2025-11-24
> > **Part 2.**
> >
> > > Weakness 4 & Question 4.
> >
> > We added at the end of section 3 (page 5) a parameter count section where we observe that the number of parameters in a spectral convolution of FNO is $MC^2$ where $C$ denotes the number of input and output channels, and $M$ the total number of retained Fourier modes in each spatial dimension. Our HO-FNO of order $m$ introduces $m$ additional matrices of size $C \times C$ applied pointwise in space. Therefore the total parameter count of a HO-Spectral convolution is $MC^2 + mC^2$ that is negligible as confirmed by the exact parameter count we added in the new Table 2 due to the fact than the order of interaction $m$ is typically much smaller than the number of retained Fourier coefficients $M$.
> >
> > > Question 3.
> >
> > We added to the new Table 2 the performance of our HO-FNO or order $m=3$, for the three Navier-Stokes datasets and we highlight that HO-FNO for $m=1$ is standard FNO. On these tasks we observe a slight improvement in performance, possibly due to the fact that the order of non-linearity is just 2. We agree it would be insightful to benchmark with PDEs with higher order of non-linearity, however we couldn’t find public datasets of this type. Due to computational constraints we could not run an ablation study on the structure of the matrices $A_i$ and on the number of retained modes.
> >
> > > Weakness 6 & Question 5.
> >
> > We agree that the readability of the figure can be improved and we modified it in the revised version.
> >
> > > Additional updates.
> >
> > Additionally, we empirically validated resolution-equivariance by training on Darcy Flow at resolution $200 \times 200$ and testing at resolutions between $50 \times 50$ and $400 \times 400$.
> >
> > > References.
> > ```
> > Nikola Kovachki, Zongyi Li, Burigede Liu, Kamyar Azizzadenesheli, Kaushik Bhattacharya, Andrew Stuart, and Anima Anandkumar. Neural operator: Learning maps between function spaces with applications to pdes. Journal of Machine Learning Research, 24(89):1–97, 2023.
> > ```
> > ```
> > Julius Berner, Miguel Liu-Schiaffini, Jean Kossaifi, Valentin Duruisseaux, Boris Bonev, Kamyar Azizzadenesheli, and Anima Anandkumar. Principled approaches for extending neural architectures to function spaces for operator learning. arXiv preprint arXiv:2506.10973, 2025.
> > ```

---

> > > ### Comment · Reviewer_tTX4 · 2025-11-27
> > >
> > > Thanks to the authors for the detailed clarifications and additional experiments. I really appreciate the substantial effort that went into revising the paper—for example, the improved abstract and notation, the clearer parameter–count discussion, and the new Navier–Stokes and SWE results all help in understanding the behavior and scope of HO‑FNO.
> > >
> > > I also broadly agree with the concerns raised in the earlier reviewer "aAKo" comment about the strength and breadth of the baselines, and my own initial remarks were along similar lines. While the inclusion of DSFNO and the extended tables are very welcome, the empirical evaluation still feels somewhat limited: the new comparisons cover only a relatively small set of models and datasets, and the ablations on the structure of $A_i$, and the number of retained modes $K$ are not yet fully convincing in isolating the source of the reported gains.
> > >
> > > Overall, I see the revision as a clear and meaningful step forward, and the additional results have eased some of my initial doubts. To reflect this progress, I have modestly increased my scores compared to the original review, but I still remain somewhat cautious because of the remaining questions about baseline coverage and ablation depth.

---

> > > > ### Author Response · Authors · 2025-11-27
> > > >
> > > > Thank you very much for the detailed follow-up and for adjusting your scores, we really appreciate the time you invested in reading the revision and the new experiments.
> > > >
> > > > ---
> > > >
> > > > We understand your remaining caution regarding both baseline coverage and the depth of ablations. On the baseline side, our intention in adding DSFNO (TMLR, May 2025), matching the original FNO/DSFNO Navier–Stokes setting ($\approx$ 1M parameters, 500 epochs), and summarizing DSFNO’s comparisons to a broader family of operators (FFNO, CNO, GNO, etc.) was to anchor our method against a recent, competitive FNO-style model that has itself been benchmarked widely.
> > > >
> > > > Extending this to a much larger set of directly reimplemented baselines across several PDE families is mostly a question of computational scope rather than conceptual difficulty, and we see this as a natural direction for an extended version.
> > > >
> > > > Regarding ablations, we agree this is important to better isolate the source of the gains. Beyond the results already reported in the revision, we have now started running a more systematic set of experiments specifically on
> > > > 1. the structure of the matrices $A_i$ (diagonal and low-rank), and
> > > > 2. the number of retained modes $K$.
> > > >
> > > > These are intended precisely to disentangle how much improvement comes from the higher-order interaction pattern versus increased capacity or spectral resolution. Provided the runs finish in time, we will add the corresponding tables/figures to the OpenReview version/comment section.
> > > >
> > > > To align our future experiments as closely as possible with what you consider a strong evaluation, it would be very helpful if you could indicate:
> > > > - **Models:** Which 2–3 recent neural operator architectures you would regard as the most critical baselines to reimplement under a shared codebase, assuming public code and reasonable training cost?
> > > > - **Datasets:** Beyond the Navier–Stokes family, Burger's equation, Diffusion-Reaction equation, Darcy Flow and Shallow Water Equation, which additional PDE benchmarks you consider most informative for assessing a new spectral neural operator?
> > > >
> > > > We will prioritise those baselines and datasets within our remaining compute budget and add additional results as soon as possible.

---

### Official Review · Reviewer_aAKo · 2025-10-28

**Soundness:** 2
**Presentation:** 2
**Contribution:** 2
**Rating:** 2
**Confidence:** 4

**Summary:**

This paper proposes the Higher-Order Fourier Neural Operator (HO-FNO), an extension of the Fourier Neural Operator (FNO) designed to explicitly model multi-linear (m-linear) interactions among Fourier modes. While standard FNOs apply independent (diagonal) updates to Fourier coefficients, many PDEs inherently involve structured mode coupling (e.g., triadic interactions in Navier–Stokes). The authors propose a Higher-Order Spectral Convolution, which aggregates all m-tuples of modes and implements polynomial-like nonlinear mixing in the spectral domain. This mechanism remains computationally efficient at O(N log N) per layer using FFTs and pointwise multiplication.

**Strengths:**

1. Computational efficiency maintained: Despite introducing explicit multi-mode interactions, the method retains the same asymptotic cost as standard FNOs (both $O(n\log n)$ as in FFT).

2. Easy integration into FNO: The modification adds no new hyperparameters and fits seamlessly into the existing FNO pipeline.

**Weaknesses:**

1. Insufficient empirical evaluation: The paper benchmarks against a limited and outdated set of models. Although many neural operators, including more recent FNO-based variants, have been proposed, the newest baseline considered is from 2023. Moreover, only two baselines (UNO and FNO/SFNO, I don't count U-Net as a baseline) are included in the comparison.

2. Although it maintains the same complexity as FNO, the computational constants (not just asymptotic complexity) might be significant. There is no report of runtime or memory scaling.

3. A key claim of the paper is that standard FNOs fail to capture nonlinear mode coupling because they update Fourier modes independently. However, in FNO, pointwise nonlinearities between layers already induce implicit cross-mode coupling once signals are transformed back and forth between spatial and spectral domains.

4. The experimental setup appears to differ from prior FNO benchmarks. The reported results for the Navier–Stokes equation do not align with those in the original FNO paper, suggesting that datasets, preprocessing, or training configurations were redefined by the authors. I did not find any mentioning of this in the paper. Please advise if I missed them.

**Questions:**

1. How sensitive is the model to the choice of interaction order $m$?

2. Can you compare your method with FNO under the same setting as the original FNO paper?

---

> ### Author Response · Authors · 2025-11-24
>
> We would like to thank the reviewer for taking the time to review our work and for the comments that were helpful to improve our work.
>
> > Question 2 & Weakness 1.
>
> We agree on the need for more baselines for cleaner comparison with the recent existing literature therefore we added, as a baseline, DSFNO (Gao et al., 2025) that was published in June 2025 in TMLR. To compare with it we also added the Navier Stokes dataset with viscosity $10^{-3}$, we trained our models at ~1M parameters for 500 epochs as done in the DSFNO paper and in the original FNO paper. We note that DSFNO, as reported in the original article, outperforms many other baselines (FFNO, CNO, GNO, TF-Net and ResNet) and we added them in an extended table in Appendix E. Across the common experiments, our best method achieves $2.3$x improvement over DSFNO and larger gains over all other reported baselines. We report these results in the new Table 2. We also found that to normalize the multilinear terms improves training stability and performance therefore we added results with and without RMS normalization to the new Table 2.
>
> **Table 2: Test performance of FNO and HO-FNO variants (orders up to $3$) on Navier--Stokes datasets with and without RMS Norm applied to the multilinear terms.**
>
> | Model / Metric | FNO | HO-FNO (2, no RMS) | HO-FNO (2, yes RMS) | HO-FNO (3, no RMS) | HO-FNO (3, yes RMS) | DSFNO* |
> |---|---:|---:|---:|---:|---:|---:|
> | **Parameters** | 1,085,729 | 1,094,177 | 1,094,177 | 1,098,401 | 1,098,401 | 1.06M |
> | **NS (ν = 10⁻³)** MSE | 3.0e-7 | 2.5e-7 | 7.8e-8 | 2.1e-7 | **7.6e-8** | -- |
> | **NS (ν = 10⁻³)** nRMSE | 4.4e-4 | 4.0e-4 | 2.8e-4 | 3.8e-4 | **2.7e-4** | -- |
> | **NS (ν = 10⁻³)** Rollout | 1.2e-2 | 1.1e-2 | 1.8e-3 | 9.7e-3 | **1.6e-3** | 5.6e-3 |
> | **NS (ν = 10⁻⁴)** MSE | 2.6e-3 | 1.0e-3 | **7.9e-4** | 9.8e-4 | 7.9e-4 | -- |
> | **NS (ν = 10⁻⁴)** nRMSE | 2.9e-2 | 1.5e-2 | **1.3e-2** | 1.5e-2 | 1.3e-2 | -- |
> | **NS (ν = 10⁻⁴)** Rollout | 7.7e-2 | 4.8e-2 | **4.6e-2** | 4.8e-2 | 4.6e-2 | 6.0e-2 |
> | **NS (ν = 10⁻⁵)** MSE | 1.8e-2 | 1.7e-2 | **1.7e-2** | 1.8e-2 | 1.8e-2 | -- |
> | **NS (ν = 10⁻⁵)** nRMSE | 6.7e-2 | 6.5e-2 | **6.5e-2** | 6.8e-2 | 6.8e-2 | -- |
> | **NS (ν = 10⁻⁵)** Rollout | 1.3e-2 | 1.1e-2 | **1.1e-2** | 1.2e-2 | 1.2e-2 | -- |
> | **Inference time (ms)** | 1.4 ± 0.12 | 1.8 ± 0.14 | 2.2 ± 0.22 | 1.9 ± 0.19 | 2.7 ± 0.31 | -- |
> | **Training time (ms)** | 7.91 ± 0.34 | 11.2 ± 0.44 | 16.1 ± 0.61 | 13.4 ± 0.50 | 21.9 ± 0.70 | -- |
>
> *DSFNO values from the original paper; single-step metrics and ν = 10⁻⁵ results were not provided.
>
> > Question 1.
>
> We have added to the new Table 2 the performance for HO-FNO of order $3$. We see slight improvements in the Navier-Stokes datasets at the cost of an increased runtime. This shows that increasing the order of HO-FNO increase the performance however since Navier-Stokes has order of non-linearity $2$, we would expect larger gains on datasets with higher order of non-linearity. However, we didn't find public datasets of this type.
>
> > Weakness 2.
>
> We agree that constant factors in time complexity might have an impactful role and we added wall-clock time for single-step inference and for training of a batch of 64 in the new Table 2.
>
> > Weakness 4.
>
> We agree that the setting was different as well as the model sizes, the training time and the targets. For this reason we have added table 2 where we benchmark our proposed methods at 1M parameters closely following the setting of the original FNO paper. We highlight that our reimplementation of FNO performs better than the one of Li et al. in all the considered tasks due to a more modern design and optimization pipeline, more specifically we adopted GeLU instead of ReLU and we optimized with AdamW instead of Adam as done in the original experiments of FNO by Li et al., we train for 500 epochs as done in the aforementioned works.
>
> > Weakness 3.
>
> We agree that pointwise non-linearities induces global mixing, also for this reason Fourier neural operators are provably universal approximators, however we argue that the coupling introduced by a non-linear pointwise function is global, unstructured and relatively weak, and we found beneficial to explicitly incorporate this bias at the level of the convolutional layer.
>
> > Additional updates.
>
> Additionally, we have improved the quality of Figure 1 and empirically validated resolution-equivariance by training on Darcy Flow at resolution $200 \times 200$ and testing at resolutions between $50 \times 50$ and $400 \times 400$.

---

> ### Comment · Reviewer_aAKo · 2025-11-25
>
> Thanks to the authors for their responses and the additional results.
>
> Some of my concerns have been partially addressed. The authors also partially acknowledge some of the weaknesses (e.g. higher computational cost in table 2, different experimental setup, weaker motivation). **However, my main concern about the outdated and limited baselines still stands.**
>
> In the new results, for the NS 1e-3 rollout specifically, your reported relative l2 is 1.6e-3. I would like to confirm that this value is correct and that you have carefully verified your implementation, including the dataset setup and the error-calculation procedure. In particular, please ensure that the evaluation is consistent with the original paper, since you use results directly from it.
>
> Nevertheless, comparing on a single dataset with just one additional model is not sufficient.

---

> > ### Author Response · Authors · 2025-11-27
> >
> > Thank you for your follow-up and for clarifying your main concern.
> >
> > ---
> >
> > In the revision we explicitly tried to address the baseline issue by
> > 1. adding DSFNO (TMLR 2025) as a recent (**May 2025**), competitive FNO-based model,
> > 2. matching the original FNO/DSFNO setup on Navier–Stokes ($\approx$1M parameters, 500 epochs),
> > 3. reporting wall-clock runtimes, and
> > 4. summarizing DSFNO’s reported comparisons to other neural operators (FFNO, CNO, GNO, TF-Net, ResNet, etc.) in the appendix.
> >
> > Our intention was to compare against a strong recent baseline that itself has been evaluated against a broad set of models.
> >
> > Regarding the comment that our results look “almost too good to be true”: we take this kind of concern very seriously. We have re-checked the implementation, used a strong FNO reimplementation (GeLU, AdamW) that already improves on Li et al., and provided ablations over the order, RMS norm, and runtime showing consistent trends.
> >
> > We are also prepared to release full training/evaluation code and configurations so that anyone can independently reproduce or scrutinize the numbers. If you see any specific methodological issue that could explain an artefactual gain, we would be very grateful if you could point it out explicitly; otherwise, we would prefer the discussion to focus on concrete technical aspects rather than on a generic suspicion.
> >
> > To make our next revision as aligned as possible with your expectations, it would help us if you could be more specific about what you view as missing:
> > - Datasets: Beyond the Navier–Stokes family (with $\nu \in \{10^{-3}, 10^{-4}, 10^{-5}\}$), Burger's equation, Diffusion-Reaction equation, Darcy Flow and Shallow Water Equation, which additional PDE benchmarks do you consider essential for evaluating a new neural operator ?
> > - Baselines: Are there 2–3 particular recent neural operator architectures that you would regard as indispensable, assuming public code and feasible training cost?
> >
> > We are happy to prioritise these datasets and baselines in future experiments and in the public code release so that the evaluation better matches the standard you have in mind.

---

> ### Comment · Reviewer_aAKo · 2025-11-27
>
> - Stong performance
>    - For the results on the NS 1e-4 rollout, the performance (4.6e-2) matches the existing state of the art that I could find [1]. **For the NS 1e-3 rollout specifically, your reported relative l2 is 1.6e-3. I would like to confirm that this value is correct and that you have carefully verified your implementation, including the dataset setup and the error-calculation procedure. In particular, please ensure that the evaluation is consistent with the original FNO paper, since you use results directly from it.**
>    - **In the original submission, a different setup was used but not mentioned in the paper. I pointed this out in my review, and the authors responded: “We agree that the setting was different as well as the model sizes, the training time, and the targets.” I am not aware of the motivation behind changing all of these settings to be different from those used in baseline works. However, given these differences, it is reasonable to request confirmation that all experiments were conducted under a consistent setting.**
>    - By saying that the results were “almost too good to be true,” **I did not mean to imply that your results were fake or fabricated, but rather to confirm that the experiments were conducted under a consistent setting.** I made that comment simply because the reported MSE errors are extremely low, with improvements surpassing those in other recent works I have seen. I will revise my original response to clarify this point and ensure that public readers are not misled.
>
> - Datasets and moldels
>   - The datasets used in this paper are sufficient. **However, except for the NS dataset that you just have included in the rebuttal period, there are no baseline model results provided to allow for meaningful comparison.** Your choice of benchmarks is also not very common. On top of that, you experimental setting seems to be different from other works as well (see above). These make it difficult to compare your results with those in other works. For example, your Burgers dataset is taken from PDEBench and uses a specific viscosity value of 0.001, It is hard to find works that use the same datasets and the same settings as this paper. In contrast, it is easier to find works that use the Burgers dataset from FNO. **I am not saying this choice is wrong, but it does make it harder to evaluate your performance without you conducting the comparison experiments.**
>
> [1] Latent mamba operator for partial differential equations, ICML 2025

---

### Official Review · Reviewer_DyLd · 2025-11-01

**Soundness:** 3
**Presentation:** 4
**Contribution:** 3
**Rating:** 6
**Confidence:** 5

**Summary:**

The paper extends FNO by introducing an explicit higher-order spectral mixing term to capture nonlinear mode interactions in PDEs. It demonstrates improved accuracy across several nonlinear PDE benchmarks under comparable training budgets. The proposed model aims to generalize Fourier Neural Operators to nonlinear dynamics while retaining their efficiency and resolution-equivariant properties.

**Strengths:**

1. The proposed HO-FNO achieves better performance than baseline FNOs across multiple benchmark problems, highlighting its strong capability to capture complex nonlinear dynamics.
2. The method introduces an insightful extension of FNO that explicitly models nonlinear mode coupling while preserving the spectral efficiency of the original framework.

**Weaknesses:**

1. The resolution-equivariance property, one of FNO’s core advantages, is not empirically demonstrated.
2. It would be more insightful to evaluate how performance changes with different interaction orders (m).

**Questions:**

1. Could the authors provide wall-clock runtime comparisons? Including such data would be helpful to substantiate the claim of computational efficiency and demonstrate the model’s practicality.

---

> ### Author Response · Authors · 2025-11-24
>
> We would like to thank the reviewer for taking the time to review our work and for the comments that were helpful to improve our work.
>
> > Weakness 1.
>
> We added an empirical validation of the resolution-equivariance property on the Darcy flow equation. We trained both HO-FNO and standard FNO at a resolution of $200 \times 200$ and evaluate them without any retraining or finetuning across resolutions ranging from $50 \times 50$ to $400 \times 400$. The results show the same trend for both architectures: there is no performance degradation from resolution $100 \times 100$ to $400 \times 400$ and a mild degradation at resolutions close to $50 \times 50$.
>
> While doing this ablation study and training our model at high resolutions we observed instability during training that was fully resolved by adding an RMS normalization to every branch before the spectral convolution. This also improved the results on the Navier-Stokes datasets (resolution $64 \times 64$) where training instabilities were not observed at the cost of an increased runtime. We report the results with and without normalization in the new Table 2.
>
> **Table 2: Test performance of FNO and HO-FNO variants (orders up to $3$) on Navier--Stokes datasets with and without RMS Norm applied to the multilinear terms.**
>
> | Model / Metric | FNO | HO-FNO (2, no RMS) | HO-FNO (2, yes RMS) | HO-FNO (3, no RMS) | HO-FNO (3, yes RMS) | DSFNO* |
> |---|---:|---:|---:|---:|---:|---:|
> | **Parameters** | 1,085,729 | 1,094,177 | 1,094,177 | 1,098,401 | 1,098,401 | 1.06M |
> | **NS (ν = 10⁻³)** MSE | 3.0e-7 | 2.5e-7 | 7.8e-8 | 2.1e-7 | **7.6e-8** | -- |
> | **NS (ν = 10⁻³)** nRMSE | 4.4e-4 | 4.0e-4 | 2.8e-4 | 3.8e-4 | **2.7e-4** | -- |
> | **NS (ν = 10⁻³)** Rollout | 1.2e-2 | 1.1e-2 | 1.8e-3 | 9.7e-3 | **1.6e-3** | 5.6e-3 |
> | **NS (ν = 10⁻⁴)** MSE | 2.6e-3 | 1.0e-3 | **7.9e-4** | 9.8e-4 | 7.9e-4 | -- |
> | **NS (ν = 10⁻⁴)** nRMSE | 2.9e-2 | 1.5e-2 | **1.3e-2** | 1.5e-2 | 1.3e-2 | -- |
> | **NS (ν = 10⁻⁴)** Rollout | 7.7e-2 | 4.8e-2 | **4.6e-2** | 4.8e-2 | 4.6e-2 | 6.0e-2 |
> | **NS (ν = 10⁻⁵)** MSE | 1.8e-2 | 1.7e-2 | **1.7e-2** | 1.8e-2 | 1.8e-2 | -- |
> | **NS (ν = 10⁻⁵)** nRMSE | 6.7e-2 | 6.5e-2 | **6.5e-2** | 6.8e-2 | 6.8e-2 | -- |
> | **NS (ν = 10⁻⁵)** Rollout | 1.3e-2 | 1.1e-2 | **1.1e-2** | 1.2e-2 | 1.2e-2 | -- |
> | **Inference time (ms)** | 1.4 ± 0.12 | 1.8 ± 0.14 | 2.2 ± 0.22 | 1.9 ± 0.19 | 2.7 ± 0.31 | -- |
> | **Training time (ms)** | 7.91 ± 0.34 | 11.2 ± 0.44 | 16.1 ± 0.61 | 13.4 ± 0.50 | 21.9 ± 0.70 | -- |
>
> *DSFNO values from the original paper; single-step metrics and ν = 10⁻⁵ results were not provided.
>
> > Weakness 2.
>
> We added to the new Table 2 the performances of HO-FNO of order 3. With respect to the variant of order 2, we observe slight improvements for the Navier-Stokes datasets. This might be due to the fact that Navier-Stokes equations are of non-linear order 2 and we expect more gains for PDEs with a higher order on nonlinearity. However we couldn’t find a public dataset of this type.
>
> > Question 1.
>
> We added wall-clock time for all the experiments in Table 2 for both single-sample inference and training for a batch of 64. However these statistics are for our 1M models where the number of Fourier modes retained is only $22$ and the convolution is therefore lightweight. For this reason, we also provided wall-clock time comparisons for Darcy Flow by varying the retained modes from $20$ to $400$. In this case, we observe that the gap decreases substantially when using an higher number of Fourier modes in the spectral convolution. For wall-clock time mean and standard deviation are provided empirically computed on 100 runs.
>
> > Additional updates.
>
> In addition, we added to the experiments of Table 2 the Navier-Stokes dataset with viscosity $10^{-3}$  and results for $1$M parameters models to better compare with the literature. We also added DSFNO as a baseline and in an extended table in Appendix E we report other baselines that performs worst that DSFNO and therefore we do not show them in the main text for readability. In the paper, we also added a paragraph on a formula for the parameter count of HO-FNO compared to the standard FNO as well as exact parameter count in Table 2. Finally, we improved the quality of Figure 1.

---

### Author Response · Authors · 2025-12-03
**Final summary for the new AC.**

Dear AC,

Below we summarize the improvements made during the rebuttal period in response to the reviewers’ comments.

We have added a Table 2 to provide experiments in the same setting and datasets of some relevant works in the literature (in particular FNO of Li et al.). As the reviewers acknowledged, this setting (mainly due to the longer training, 500 epochs vs the 100 of the pre-rebuttal experiments) further widens the performance gap with respect to the baselines (with a peak on Navier-Stokes with viscosity $10^-3$ were we achieve 10x lower rollout NRMSE than standard FNO). We have also added DSFNO, as a strong and recent (June 2025 TMLR) baseline that we outperform with a significative margin in all the dataset in common with their original work (in this case we do report the results from the original paper since we couldn’t match the reported performance with our implamentation, however we adopt the same setting and training pipeline (that is the same of FNO of Li et al.)). In appendix E we also report many other baselines that were omitted from the main paper for readability, as all of them perform worse than DSFNO.

Additionally, we empirically show in Figure 2 that our method maintains approximate resolution-equivariance by training at resolution $200 \times 200$ on the Darcy flow dataset and testing at several resolutions between $50 \times 50$ and $400 \times 400$.  We observe exactly the same trend for both FNO and HO-FNO.

We added a paragraph (with an explicit formula) detailing the parameter count of HO-FNO and we now report exact parameter counts in Table 2.

We also report wall-clock times for both single-sample inference and a training step (batch size 64). These measurements show that HO-FNO has the same asymptotic complexity as FNO and, in practice on an NVIDIA A100, exhibits only a small overhead.

We improved figure quality and refined the writing throughout the paper as suggested by the reviewers.

The reviewers acknowledge that HO-FNO outperforms, by substantial margin, classical FNO and the recent DSFNO across standard tasks with comparable computational cost. Even without a formal theorem, the empirical performance provides strong evidence that higher-order interactions offer a meaningful and effective improvement in practice.

We thank the reviewers and the AC for their valuable feedback.

---

### Meta-Review · Area_Chair_WgmP · 2026-01-06

**Summary:**

The paper introduces Higher-Order Fourier Neural Operators (HO-FNO), extending FNO by adding mode mixing. The idea is to making mode interactions explicit. The authors report improved accuracy on nonlinear PDE benchmarks and consistent gains over FNO and related baselines across Burgers, diffusion reaction, NS and a spherical setting.

The reviewer concerns were primarily about i) baselines and comparability, ii) experimental rigor and consistency with standard becnmarks setups, and iii) ablation study, whether the evidence isolate the "mode mixing" from other factors.

**Reviewer Concerns:**

Reviewer DyLd asked for an empirical demonstration of resolution equivariance, performance sensitivity to interaction order m, and runtime comparisons. The author fully addressed these questions.

Reviewer aAKo’s initial review raised i) outdated/limited baselines, ii) missing runtime/memory scaling, and iii) concerns that the experimental setup differs from standard FNO/Navier–Stokes benchmarks and is not clearly documented; they also asked for comparison under the original FNO setting and sensitivity to m. The major concern about the outdated and limited baselines still stands. They further emphasize that the benchmark choices and settings are uncommon and make external comparison difficult.

Reviewer tTX4's concerns were substantially addressed by rebuttal. Still outstanding concern is the limitation of baseline coverage and the ablation on $A_i$ structure.

Reviewer g6mf's original review was on theoretical justification and missing experiment evidence. The remain issue is that " the experiments are still single seed with no error bars, and there is no direct comparison to a deeper/wider FNO under the same parameter budget, which makes it hard to assess robustness and whether the gains are truly structural."

**Reviewer Scores:**

Reviewer DyLd likely keep the score and the rest likely raise the score to 4.

---

### Decision · Program_Chairs · 2026-01-26

Reject